# ADAM: AN EMBODIED CAUSAL AGENT IN OPEN-WORLD ENVIRONMENTS

**Shu Yu[1,2,3], Chaochao Lu[1,2†]**
[1]Shanghai Artificial Intelligence Laboratory  [2]Shanghai Innovation Institute  [3]Fudan University
{yushu,luchaochao}@pjlab.org.cn

## ABSTRACT

In open-world environments like Minecraft, existing agents face challenges in continuously learning structured knowledge, particularly causality. These challenges stem from the opacity inherent in black-box models and an excessive reliance on prior knowledge during training, which impair their interpretability and generalization capability. To this end, we introduce ADAM, An emboDied causal Agent in Minecraft, which can autonomously navigate the open world, perceive multimodal context, learn causal world knowledge, and tackle complex tasks through lifelong learning. ADAM is empowered by four key components: 1) an interaction module, enabling the agent to execute actions while recording the interaction processes; 2) a causal model module, tasked with constructing an ever-growing causal graph from scratch, which enhances interpretability and reduces reliance on prior knowledge; 3) a controller module, comprising a planner, an actor, and a memory pool, using the learned causal graph to accomplish tasks; 4) a perception module, powered by multimodal large language models, enabling ADAM to perceive like a human player. Extensive experiments show that ADAM constructs a nearly perfect causal graph from scratch, enabling efficient task decomposition and execution with strong interpretability. Notably, in the modified Minecraft game where no prior knowledge is available, ADAM excels with remarkable robustness and generalization capability. ADAM pioneers a novel paradigm that integrates causal methods and embodied agents synergistically. Our project page is at https://opencausalab.github.io/ADAM.

## 1 INTRODUCTION

Embodied agents exploring open-world environments mark a critical frontier in artificial intelligence (AI) research (Cassell, 2000; Xia et al., 2018; Savva et al., 2019). The ultimate goal is to build generally capable agents (GCAs) (Team et al., 2021) that can autonomously perform a broad range of tasks through perception, learning, and interaction (Mnih et al., 2015; Xi et al., 2023; Park et al., 2023). Minecraft (Nebel et al., 2016) is a globally renowned 3D video game, where players need to master complex crafting recipes (*e.g.*, planks 🟫 + sticks ╱ → wood_pickaxe ⛏) and gather resources (*e.g.*, mining cobblestone 🪨 with wood_pickaxe ⛏), progressively unlocking new items in the technology tree. The substantial freedom and precise simulation of physical laws in Minecraft render it an exceptional platform for researching GCAs.

In Minecraft, two primary approaches for developing GCAs have been extensively explored: reinforcement learning (RL)-based (Lin et al., 2021; Baker et al., 2022; Fan et al., 2022; Mao et al., 2022; Hafner et al., 2023) and large language model (LLM)-based (Wang et al., 2023a; Zhu et al., 2023; Qin et al., 2023a; Nottingham et al., 2023; Wang et al., 2023c;d). Specifically, RL agents learn through interactions and updating their black-box model weights, which poses challenges for interpretability, efficiency, and generalization. On the other hand, LLM-based agents possess and rely on rich prior knowledge of both virtual games and real world (Ouyang et al., 2022; Wei et al., 2022a; Achiam et al., 2023). Their reliance on omniscient data (*e.g.*, GPS coordinates, voxel blocks, biome, *etc.*, which are not explicitly observable by the player) (Wang et al., 2023a; Zhu et al., 2023; Wang et al., 2023d) presents challenges for generalization and human gameplay observation alignment.

---

†Corresponding author.

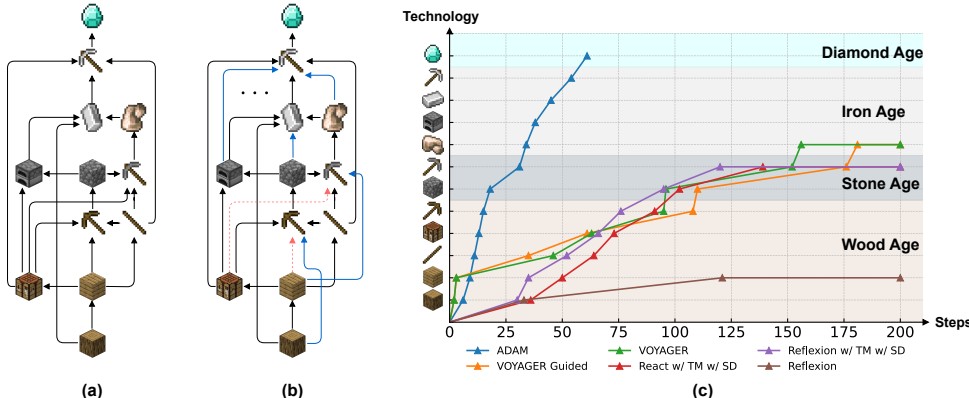

Figure 1: (a) The technology tree for acquiring `diamonds` 💎 in the Minecraft game. ADAM can precisely discover item dependencies from scratch. (b) Modified Minecraft technology tree, where the prior knowledge from the Internet or Wiki **does not align with** the actual game dynamics. Red arrows denote removed dependencies, while blue arrows denote added dependencies. (c) In the game setting shown in (b), ADAM maintains the ability to learn the correct causal graph and successfully obtains `diamonds` 💎, whereas other methods can only acquire `raw_iron` 🪨 within the step limit, and ADAM achieves a 4.6× speedup in obtaining `raw_iron` 🪨 compared to the SOTA.

To address these issues, we propose ADAM, An emboDied causal Agent in Minecraft, which can autonomously navigate the open world, perceive multimodal context, learn causal world knowledge, and tackle complex tasks through lifelong learning. Specifically, as shown in Fig. 2, ADAM is composed of four key modules: (1) **Interaction module**, which enables the agent to execute actions from the action space and processes the agent's observable information into formatted records. (2) **Causal model module**, which includes two causal discovery (CD) methods. LLM-based CD utilizes interaction records to make causal assumptions. Intervention-based CD refines these assumptions to derive a causal subgraph. Multiple causal subgraphs are integrated into a comprehensive causal graph (*i.e.*, technology tree). (3) **Controller module**, which includes a planner, an actor, and a memory pool. The planner can utilize the causal graph to perform task decomposition. The actor uses the decomposed subtasks for action choosing. The memory pool ensures the long-term context dependence. (4) **Perception module**, which is driven by multimodal LLMs (MLLMs), enabling ADAM to perceive its surroundings without relying on omniscient data, thereby achieving human-like gameplay observation.

Extensive experiments demonstrate that ADAM achieves a 2.2× speedup compared to the SOTA in the task of obtaining `diamonds` 💎. In scenarios where crafting recipes are modified (Fig. 1b), only ADAM maintains the ability to obtain `diamonds` 💎, while other methods can only acquire `raw_iron` 🪨 within the step limit, and ADAM achieves a 4.6× speedup in obtaining `raw_iron` 🪨 compared to the SOTA (Fig. 1c). ADAM demonstrates strong interpretability through constructing a nearly perfect technology tree (Fig. 1a) from scratch, whereas other methods exhibit at least 30% errors or omissions. Meanwhile, in tasks requiring environmental perception, ADAM closely aligns with human gameplay observation without relying on omniscient metadata, while maintaining comparable performance to methods that utilize such data.

Overall, our contributions are as follows:

(1) **We introduce ADAM, an embodied causal agent** that can autonomously navigate the open world, perceive multimodal context, learn causal world knowledge, and tackle complex tasks through lifelong learning.

(2) **We address the limitations of existing embodied agents.** Our ADAM demonstrates strong robustness without relying on prior knowledge or omniscient metadata used by other LLM-based agents, while aligns human gameplay observation during exploration.

(3) **We pioneer the integration of causal methods into open-world embodied agents**, allowing the agent to organize the learned knowledge into a rigorous causal graph, thereby demonstrating excellent interpretability.

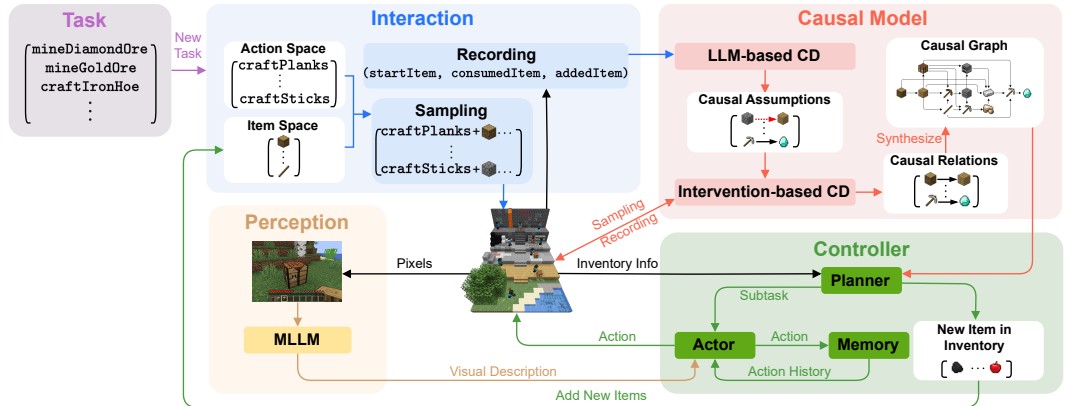

Figure 2: Four key modules of ADAM. The **interaction module** executes actions in the environment according to the task and records the processes. The **causal model module** identifies the causal relationship between items and actions to construct an ever-growing causal graph. The **controller module** implements task execution based on the learned causal graph. The **perception module** aligns the agent's gameplay observation more closely with human.

(4) **We improve the CD performance by employing embodied agent-driven interventions**, which enhances the accuracy and efficiency of CD compared to existing methods without interventions.

## 2 PRELIMINARIES

**Causal graphical models (CGMs).** A CGM represents the structure of causality within a system (Peters et al., 2017) by detailing the direct causal relationships among a set of variables $X_1, \ldots, X_n$. It is characterized by a distribution over these variables and is associated with a directed acyclic graph (DAG), known as a causal graph. In this graph, each node corresponds to a variable, and each directed edge signifies a direct causal relation from $X_i$ to $X_j$.

**Causal discovery from interventions.** Causal Discovery (CD) (Spirtes et al., 2001; Pearl, 2009; Peters et al., 2017; Glymour et al., 2019) is a fundamental process to infer causal relationships from data. The relationships are typically represented in the form of a causal graph. While observational data reveals correlations, interventions allow us to analyze causal dependencies between variables. Specifically, interventions alter the distribution of variables (*e.g.*, the initial system state) during experimental sampling, which serves as a gold standard for CD (Eberhardt & Scheines, 2007). By observing their effects, we can identify causal relationships rather than merely correlations.

## 3 METHOD

In this section, we begin with the basic notations and definitions in our work (Sec. 3.1). Then, we give an overview of our ADAM framework (Sec. 3.2). Next, we detail the four modules of ADAM: interaction module (Sec. 3.3), causal model module (Sec. 3.4), controller module (Sec. 3.5), and perception module (Sec. 3.6).

### 3.1 NOTATIONS AND DEFINITIONS

We first introduce several key notations and definitions in our work. Sets are denoted by uppercase letters, and their elements by lowercase letters.

**Inventory**: The set $I_t$ of items possessed by the agent at any step $t$. **Initialization**: At the initialization of Minecraft instances, the initial inventory $I_0$ can be specified by the agent. **Action Space**: The set $A$ of actions whose names (*e.g.*, `gatherWoodLog`) are replaced by letters and invisible.

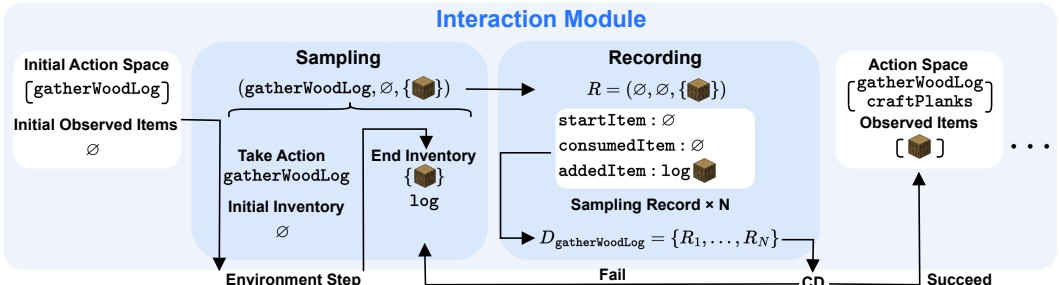

Figure 3: The interaction module has two core functionalities: **sampling** and **recording**. Sampling involves executing actions in the environment, and recording involves processing and documenting the observable information. For instance, the initial action space is {gatherWoodLog}, whose name is not exposed to ADAM and is denoted as {$a$} here (Note that, the original notation {gatherWoodLog} is retained in the figure for the illustrative purpose.). The initial observed item space is $\varnothing$. After executing $a$ for one step, logs (🪵) are obtained. A sampling can be represented as $(a, \varnothing, \{🪵\})$, where $\varnothing$ is the initial inventory and $\{🪵\}$ is the inventory after this step. The result is recorded as $R = (\varnothing, \varnothing, \{🪵\})$, where the first $\varnothing$ is the initial inventory and the second $\varnothing$ indicates that no items are consumed, and $\{🪵\}$ represents the items that are obtained. After sampling $N$ times, data $D_a = \{R_1, \ldots, R_N\}$ is provided to the causal model module for CD. If the causal relation failed to be identified, resampling on $a$ occurs; if successful, new actions like craftPlanks are enabled by the acquisition of 🪵, and the observed item space is updated to {🪵}.

ADAM must independently discover the effects of these actions. **Movement Space**: The set $M$ of basic movements whose names (*e.g.*, moveForward, moveBackward) are visible to ADAM. **Step**: The agent takes an action $a$ in the environment. A step ends either when action completes or when execution times out. **Observed Item Space**: The set $S$ of all items that ADAM has encountered. Initially, $S$ is empty. **Environmental Factors**: The set $\mathcal{E}$ of environment conditions (*e.g.*, biome, surrounding block types). **Task**: Denoted by the tuple $(I_{\text{goal}}, \mathcal{E})$, a task is accomplished at step $t$ if $I_{\text{goal}} \subseteq I_t$ and the factors $\mathcal{E}$ are present within a certain distance around the agent.

## 3.2 OVERVIEW

ADAM comprises four modules as depicted in Fig. 2, which can be extended to other open-world environments, as discussed in Appendix F. Given a task $(I_{\text{goal}}, \mathcal{E})$, the **interaction module** enables the agent to execute each action $a$ and records data $D_a$. This data is then utilized by the **causal model module**, which employs LLM-based CD to make causal assumptions and intervention-based CD to refine these assumptions into causal subgraphs. These subgraphs are integrated into a causal graph (*i.e.*, technology tree). Once the causal graph $\mathcal{G}$ contains all required items $I_{\text{goal}}$ in the task, the **controller module** executes the task from an empty inventory, aided by visual descriptions from the **perception module**. Newly discovered items enable the execution of new unknown actions and the CD on these actions. This iterative process ensures the lifelong learning through continuous engagement and adaptation.

## 3.3 INTERACTION MODULE

The interaction module (Fig. 3) enables the agent to execute actions for sampling and records observable information. Initially, the action space $A$ contains one element gatherWoodLog, which is the most basic action in Minecraft and a common setup for Minecraft agents (Wang et al., 2023a). As the agent acquires certain new items, new actions are enabled in $A$. For example, the action of mining becomes available only after the agent obtains a mining tool (*e.g.*, a wooden_pickaxe 🪓). By continuously collecting data on each action and coordinating with other modules, the nodes on the technology tree are progressively discovered by the agent.

**Sampling.** This involves initiating a Minecraft instance with all observed items $S$ as the initial inventory $I_0$, and then executing action $a$ to observe the agent's inventory $I_1$ after this step. The quantity of each item in $S$ and $I_1$ is recorded. This sampling, represented as $(a, S, I_1)$, is performed

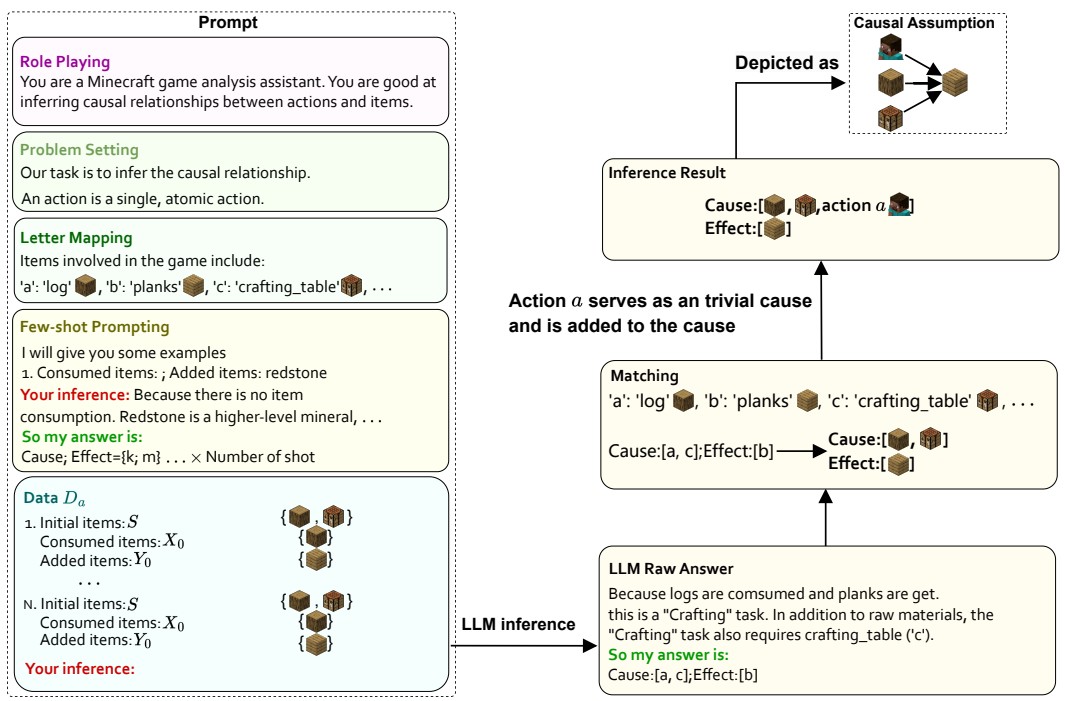

Figure 4: LLM-based CD performs causal reasoning under the guidance of the prompt. Role Playing assigns an analysis assistant role to the LLM. Problem Setting provides the reasoning task. Letter Mapping maps the item names to letters for the accurate output. Few-shot Prompting provides examples for chain-of-thought (Wei et al., 2022b) reasoning. Data $D_a$ is presented in the same form as the few-shot examples. The output of LLM serves as the causal assumption.

$N$ times, focusing on one specific action at a time. Due to the same configuration of these $N$ samplings, parallelization is available and accelerates the exploration.

**Recording.** This involves processing the results of samplings and documenting them. For the sampling $(a, S, I_1)$, **consumed items** are denoted by $X$, which includes items with decreased quantities and items that are present in $S$ but absent in $I_1$. Conversely, **obtained items** are denoted by $Y$, which includes items with increased quantities and items that newly appear in $I_1$. A single record is denoted as $R = (S, X, Y)$. Such N records on action $a$ collectively form data $D_a = \{R_1, \ldots, R_N\}$.

### 3.4 CAUSAL MODEL MODULE

The causal model module uses data $D_a$ to infer causal relationships and constructs causal subgraphs for each action $a$. These subgraphs are then integrated into a comprehensive causal graph.

A causal relationship is a stable and repeatable dependence in any step, where the agent takes an action $a$ and the acquisition of items $E$ (*effect* items) relies on the items $C$ (*cause* items) the agent possessed before the step. Since the technology tree focuses on new item acquisition, causal relations where no items are obtained do not contribute to the technology tree.

In the causal model module, **LLM-based CD** makes assumptions on the causal relationships, which effectively reduces the number of item nodes that need to be confirmed in the causal subgraph and achieves acceleration. Then, **intervention-based CD** refines these assumptions and accurately constructs causal subgraphs. We also detail our **optimization techniques** employed in this module.

**LLM-based CD.** The input to LLM-based CD (Fig. 4) is the data $D_a$ containing $N$ records, and the output is a causal assumption, which consists of cause items $C$ and effect items $E$. The prompt is designed with five components: **(1) Role Playing**, which assigns a specific role to the LLM. In this

context, the LLM serves as a causal analysis assistant, dedicated to extracting causal relationships from data. **(2) Problem Setting**, which provides the reasoning task and the fundamental concepts of the environment. We avoid introducing specific environmental knowledge for generalization. **(3) Letter Mapping**, which involves mapping item names to letters, a simplification that facilitates the formatted output. **(4) Few-shot Examples**, which involves providing the LLM with several reasoning examples in a chain-of-thought (Wei et al., 2022b) style, including example questions, the reasoning processes, and the expected answering format. The examples are independent of the technology tree, thus preventing the introduction of prior knowledge. **(5) Data** $D_a$, which consists of $N$ records from the interaction module, and is formatted consistently with the few-shot examples.

**Intervention-based CD.** Intervention is a method to experimentally verify causal relationships among variables. Intervention-based CD (Fig. 5) can refine the causal assumptions and construct a highly accurate causal graph.

Before interventions, it has to be confirmed that $C$ is a sufficient condition for $E$. If $C$ has already lacked the necessary items to achieve $E$, then the vital edges are missing and the graph can not be corrected by excluding redundant edges. Specifically, by performing sampling $(a, C, I_1)$ as described in Sec. 3.3, if $E$ is consistently absent from $I_1$, the assumption is deemed incorrect, leading the LLM-based CD to re-infer the assumption. If these inferences continue to fail, the interaction module will resample data $D_a$.

Intervention-based CD performs samplings $(a, C \setminus \{c\}, I_1)$ for each item $c \in C$. For each item $e \in E$, if there is always $e \notin I_1$ within a maximum number of samplings, then $c$ is the cause of $e$, and the edge $c \rightarrow e$ is retained. If at least one sample includes $e \in I_1$, then $c$ is not indispensable for $e$, and the edge $c \rightarrow e$ is removed. Intervention-based CD yields accurate causal subgraphs, which are subsequently integrated into a complete causal graph.

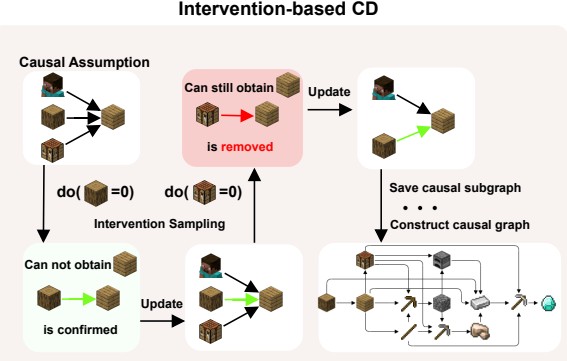

Figure 5: Intervention-based CD verifies causal assumptions. An example of causal assumption is that, under the action $a$, `log` (🪵) and `crafting_table` (🛠) contribute to the acquisition of `planks` (🟫). This assumption is denoted as $(a, \{🪵, 🛠\}, \{🟫\})$. Intervention-based CD will verify each item in $\{🪵, 🛠\}$. When removing 🪵 from the inventory and executing action $a$, 🟫 cannot be obtained, proving that 🪵 is a dependency of 🟫, and this edge (🪵 $\rightarrow$ 🟫) is retained in the causal graph (represented in green). When removing 🛠 and executing action $a$, 🟫 still can be obtained, which shows that 🛠 is not a dependency of 🟫, and this edge (🛠 $\rightarrow$ 🟫) is removed from the causal graph (represented in red). Intervention-based CD results in a corrected causal subgraph. Multiple subgraphs can be combined into the technology tree in Minecraft. The actions are not shown in the technology tree for the sake of simplicity.

**Optimization techniques.** (1) *Temporal Modeling (TM)*: Temporal information can help determine causal direction. Events occurring later cannot be the cause of events occurring earlier. This eliminates some edges in the causal graph. (2) *Subgraph Decomposition (SD)*: By focusing on the causal effects of individual actions, the causal model module processes a manageable number of items (only those in each cause-effect pair) at a time. This significantly reduces the complexity of the CD process.

## 3.5 CONTROLLER MODULE

During the execution of task $(I_{\text{goal}}, \mathcal{E})$, ADAM begins with an empty inventory (*i.e.*, $I_0 = \varnothing$) to ensure fair comparison with other methods. Once all items in $I_{\text{goal}}$ have been obtained in the causal graph, the controller module (Fig. 6) is responsible for executing the task. If there are newly discovered items, they will be added to the observed item space $S$ and pend for a new cycle of CD, thus achieving lifelong learning. The controller module comprises three components as described below.

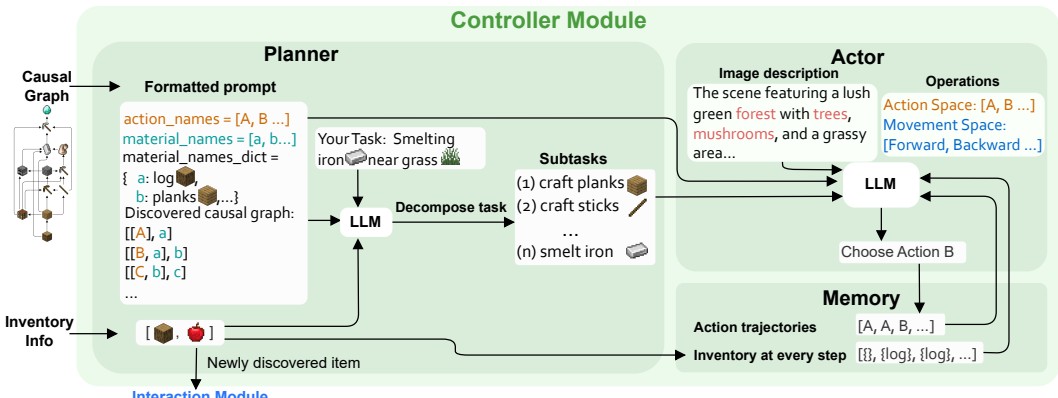

Figure 6: The controller module comprises three components. The Planner utilizes the reasoning capabilities of LLMs to decompose the task. It receives the current inventory and the learned causal graph as input. The Actor leverages LLMs to choose an action $a$ in the the action space $A$ or a movement $m$ in the movement space $M$, and executes it in the environment. The Memory records the step information, including action trajectories and item changes every step.

**Planner.** The Planner utilizes LLMs to decompose the task with current inventory $I_t$ at step $t$ and the learned causal graph. Relying solely on the causal graph is suboptimal as actions may fail or have side effects. LLMs can fully utilize the inventory information and provide detailed thought process of the decomposition process, which is passed to the Actor for action selection.

**Actor.** The Actor leverages LLMs to choose an action $a \in A$ or a movement $m \in M$ to execute. It receives the task decomposition from the Planner, the description of game image at this step, and the records from the Memory. In the task $(I_{\text{goal}}, \mathcal{E})$, the Actor prioritizes obtaining the items $I_{\text{goal}}$, because the agent's surroundings $\mathcal{E}_t$ at each step $t$ change during this process, which may affect the acquisition of $\mathcal{E}$.

**Memory.** The Memory records observable information during task execution, including action trajectories and inventory changes at each step. It plays a crucial role in tracking long-term dependencies and facilitates robust task execution.

### 3.6  PERCEPTION MODULE

The perception module utilizes MLLMs for environmental observation, enabling ADAM to perceive the world without relying on metadata such as the names of surrounding blocks, GPS coordinates, or biome names, which are typically invisible to human players. This module captures first-person screenshots between steps, which are processed by MLLMs to generate text descriptions. This text description is subsequently passed to the Actor in the controller module for action selection.

## 4  EXPERIMENTS

### 4.1  EXPERIMENTAL SETUP

In our study, we employ Mineflayer (PrismarineJS, 2023a), a JavaScript-based framework providing control APIs for the commercial Minecraft (version 1.19) [1]. The encapsulation of Mineflayer uses the implementation in VOYAGER (Wang et al., 2023a). For visual processing, we utilize prismarine-viewer (PrismarineJS, 2023b), an API for rendering game scenes from the agent's perspective. ADAM and our baselines all use GPT-4-turbo (gpt-4-0125-preview) (Achiam et al., 2023) for LLM inference, with the temperature set to 0.3 based on our experiments in Appendix A. For visual description, we utilize LLaVA-v1.5-13B (Liu et al., 2024) in our perception module.

---

[1]https://www.minecraft.net

| Model | SHD | Model | SHD |
|---|---|---|---|
| ADAM | **2 ± 2** | CDHRL | 10 ± 4 |
| ADAM w/o TM,SD | 19 ± 6 | CDHRL w/ SD | 6 ± 2 |
| Reflexion | 24 ± 9 | React w/ TM,SD | 5 ± 2 |
| AutoGPT | 24 ± 6 | Reflexion w/ TM,SD | 4 ± 2 |
| Empty Graph | 32 | AutoGPT w/ TM,SD | 4 ± 2 |

Table 1: Structural Hamming Distance (SHD) between the learned causal graph and the target graph. For non-embodied agents without built-in interventions, even with our TM and SD (Section 3.4), the causal graph learned by these agents remains suboptimal. The CD used by CDHRL is based on SDI (Ke et al., 2019), which incorporates temporal modeling (TM) into its implementation, and still exhibits over 30% errors or omission, while ADAM can identify a nearly perfect causal graph.

## 4.2 BASELINES

In the absence of directly comparable works, we select representative methods as baselines and focus on their comparable aspects, including: (1) **ReAct** (Yao et al., 2023), which explicitly expresses the thought processes through chain-of-thought prompting (Wei et al., 2022b). (2) **Reflexion** (Shinn et al., 2024), derived from ReAct, which can reflect on its exploration history. (3) **AutoGPT** (Significant Gravitas), which can autonomously decompose tasks and execute subtasks in a ReAct-style. Baselines 1–3 can only perform text-based tasks and lack embodied components to interact with the environment. Hereafter, they are referred to as **non-embodied agents**, and we have adapted them with our interaction module for embodied exploration. (4) **VOYAGER** (Wang et al., 2023a) is an LLM-based embodied lifelong learning agent in Minecraft, featuring an automatic curriculum aiming to *"discover as many diverse things as possible"*. We also add our benchmarking tasks (*e.g.*, obtaining diamonds 💎) to the curriculum for oriented explorations, denoted as **VOYAGER-Guided**. (5) **CDHRL** (Peng et al., 2022) introduces an RL agent that constructs hierarchical structures based on causal relationships. Given that RL agents have **disparate action spaces** and **magnitudes** of difference in episode length compared to LLM-based methods ($10^5 \sim 10^8$ in **DreamerV3** (Hafner et al., 2023) and **DEPS** (Wang et al., 2023d) versus $10^1 \sim 10^2$ in VOYAGER and our ADAM), we focus our comparison on the CD component of CDHRL. Detailed discussion of other Minecraft agents that are not directly comparable can be found in Appendix C.

## 4.3 MAIN RESULTS

**Interpretability.** We evaluate the interpretability of agents by assessing their ability to construct a causal graph. Structure Hamming Distance (SHD) (Zheng et al., 2024) can quantify the discrepancy between the learned causal graph and the target graph as presented in Tab. 1. Despite applying our TM and SD optimization (Section 3.4), non-embodied agents without built-in interventions fail to achieve the optimal accuracy. For CDHRL, we directly provide ADAM's sampling data for its CD. CDHRL performs CD with all nodes in the causal graph, which hampers its performance as CDHRL shows improved performance when integrated with our SD optimization. Nevertheless, these competitive methods exhibit at least 30% errors or omissions, whilst ADAM is capable of learning a **nearly perfect causal graph**. VOYAGER does not organize knowledge in a causal graph.

**Efficiency.** Efficiency is evaluated in the original Minecraft as shown in Tab. 2. ADAM achieves a **2.2× speedup** compared to the SOTA in the task of obtaining diamonds 💎. The design of parallel sampling (Section 3.3) significantly boosts ADAM's exploration efficiency. Due to the absence of intervention-based CD, non-embodied agents are unable to refine their causal assumptions, therefore limiting their exploration speed and confining them to the lower levels of the technology tree. Additionally, ADAM achieves **higher success rate** across most tasks compared to other methods.

**Robustness.** Robustness is evaluated in a modified Minecraft where crafting recipes are altered. The result is shown in Fig. 1c. In this scenario, there exists a misalignment between the LLM's prior knowledge and the actual game dynamics. ADAM successfully obtains diamonds 💎 in the modified Minecraft. The baselines lack a CD approach to learn and organize new causal knowledge beyond prior, and struggle with complex dependencies. The most advanced item that baseline agents

| Framework | Wooden Tool | Stone Tool | Iron Tool | Diamond |
|---|---|---|---|---|
| React w/ TM w/ SD | $51 \pm 19(2/3)$ | $96(1/3)$ | N/A $(0/3)$ | N/A $(0/3)$ |
| Reflexion w/ TM w/ SD | $60 \pm 27(3/3)$ | $122 \pm 56(2/3)$ | N/A $(0/3)$ | N/A $(0/3)$ |
| AutoGPT w/ TM w/ SD | $49 \pm 20(2/3)$ | $103 \pm 45(2/3)$ | N/A $(0/3)$ | N/A $(0/3)$ |
| VOYAGER | $8 \pm 2(3/3)$ | $\mathbf{10 \pm 3(3/3)}$ | $27 \pm 11(3/3)$ | $113 \pm 41(2/3)$ |
| VOYAGER Guided | $\mathbf{7 \pm 2(3/3)}$ | $11 \pm 2(3/3)$ | $\mathbf{24 \pm 9(3/3)}$ | $75 \pm 20(2/3)$ |
| ADAM | $23 \pm 5(3/3)$ | $33 \pm 8(3/3)$ | $53 \pm 16(3/3)$ | $68 \pm 21(3/3)$ |
| ADAM Parallel | $12 \pm 2(3/3)$ | $18 \pm 3(3/3)$ | $29 \pm 5(3/3)$ | $\mathbf{34 \pm 7(3/3)}$ |

Table 2: Exploration steps in different tasks. Fewer steps indicates higher efficiency. Each method has three trials for a maximum length of 200 steps. The success rate is depicted in the parentheses. ADAM achieves a $2.2\times$ speed compared to the SOTA in the task of obtaining diamonds, with a higher success rate.

| | VOYAGER | VOYAGER w/o Meta | ADAM | ADAM w/o MLLM |
|---|---|---|---|---|
| Find a river | $\mathbf{16 \pm 8(2/3)}$ | N/A $(0/3)$ | $21 \pm 16(\mathbf{2/3})$ | N/A $(0/3)$ |
| Gather log near river | $\mathbf{36}(1/3)$ | N/A $(0/3)$ | $40 \pm 23(\mathbf{2/3})$ | N/A $(0/3)$ |
| Smelting iron near grass | N/A $(0/3)$ | N/A $(0/3)$ | $\mathbf{95}(1/3)$ | N/A $(0/3)$ |

Table 3: Performance of ADAM and VOYAGER in tasks requiring environmental factors $\mathcal{E}$. Each method has three trials for a maximum length of 100 steps. The success rate is depicted in the parentheses. VOYAGER's performance significantly declines when metadata is not accessible, whereas ADAM do not rely on metadata. MLLM contributes ADAM's performance in this type of tasks.

manage to acquire is raw_iron. In terms of exploration speed, ADAM achieves a **4.6× speedup** in obtaining raw_iron. For more details, please refer to Tab. 7 in the Appendix.

**Lifelong learning.** ADAM utilizes CD methods to learn the effects of each action, obtaining causal subgraphs that contain new items. These new items make it possible to perform more unknown actions, thereby continually expanding the knowledge of the game world in a bootstrapping manner and achieves lifelong learning. ADAM successfully learns a complex causal graph of all 41 actions we implement, as demonstrated in Fig. 7.

**Human gameplay alignment.** Avoiding the use of human-invisible metadata demonstrates ADAM's alignment with human gameplay. We compare VOYAGER, a fully text-based agent that relies on metadata[2]. We test three tasks that requires environmental factors $\mathcal{E}$. The results are shown in Tab. 3. VOYAGER's performance significantly declines when metadata is not accessible. ADAM performs well on these tasks relying solely on observable information. Ablation experiments demonstrate that MLLMs contribute to ADAM's performance on this type of task.

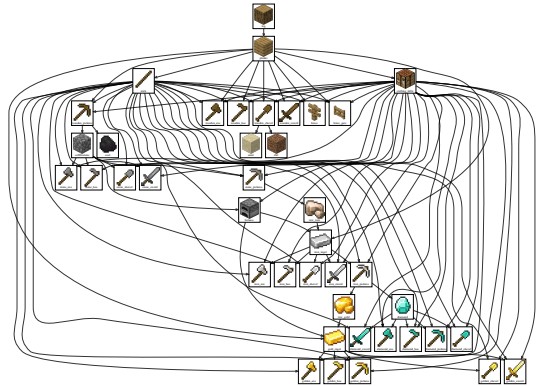

Figure 7: The causal graph learned in lifelong learning. ADAM successfully unlocks all 41 actions we implement and discovers accurate causal relationships.

### 4.4 ABLATION STUDIES

**Ablation of LLM-based CD** Prior knowledge and inference capabilities are key factors in ablating LLM-based CD. Prior knowledge can be ablated in modified games and enhanced through fine-tuning LLMs on Minecraft knowledge datasets as detailed in Appendix B. Inference capabilities can be ablated by replacing SOTA LLMs with smaller LLMs. Our result in Tab. 4 shows that ADAM primarily utilizes the reasoning abilities of LLMs rather than relying on prior knowledge.

---

[2]The information used by VOYAGER and ADAM is compared in Tab. 5 in the Appendix.

| Model | R&E (success/all) | Model | R&E (success/all) |
|---|---|---|---|
| gpt-4-turbo-preview | **0.0 (35/35)** | gpt-4 | 0.1 (35/35) |
| gpt-4-turbo-preview[†] | 0.1 (35/35) | gpt-4[†] | 0.1 (35/35) |
| gpt-3.5-turbo | 0.2 (34/35) | Llama-2-70B | 0.6 (23/35) |
| gpt-3.5-turbo[†] | 0.3 (32/35) | Llama-2-70B[†] | 0.9 (16/35) |
| Llama-2-70B-finetuned | 0.4 (27/35) | Llama-2-13B-finetuned | 1.8 (5/35) |
| Llama-2-70B-finetuned[†] | 0.9 (15/35) | Llama-2-13B-finetuned[†] | N/A (0/35) |

Table 4: Ablation of LLM-based CD. † means "in a modified environment". We record the average number of redundant/error items (represented as R&E) in the causal assumption proposed by LLM-based CD, and the success rate after the intervention-based CD. LLMs with only strong prior knowledge but weak inference abilities cannot perform well.

**Ablation of intervention-based CD.** Without intervention-based CD, agents are forced to rely on exhaustive trials to learn the game knowledge, significantly impairing their efficiency and effectiveness. Our experimental results are shown in Fig. 8. Through interventions, ADAM achieves up to 4.4 × speed and a higher accuracy compared to the ablated group.

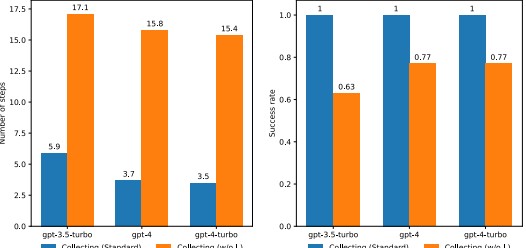

Figure 8: Average number of steps and success rate to learn the causal subgraphs of "Collecting" actions (*e.g.*, `gatherIronOre`), which are representative due to higher noise in their sampling data. We perform up to 20 steps for each action, and if the CD fails, it is counted as 20. Intervention contributes up to 4.4× acceleration and higher success rate in the exploration.

## 5 RELATED WORK

**Causality in Agents.** The integration of causality (Pearl, 2009; Peters et al., 2017; Schölkopf, 2022) into agents is primarily aimed at enhancing the learning efficiency (Méndez-Molina et al., 2020; Seitzer et al., 2021; Gasse et al., 2021; Sun et al., 2021; Peng et al., 2022). Peng et al. (2022) propose CDHRL to build high-quality hierarchical structures in complicated environments. Méndez-Molina et al. (2020) employ the causal models to restrict the search space. Zeng et al. (2023) distinguish agents with causality in two categories: ones relying on prior causal information and ones that learn causality by causal discovery algorithms (Spirtes et al., 2000; Sun et al., 2007; Zhang & Hyvärinen, 2009; Zhang et al., 2011; Peters et al., 2014; Zhu et al., 2019). Our work aligns with the latter category and extends to a wider range of scenarios where prior knowledge is unknown or not available.

**LLM/MLLM-Based Agents.** Leveraging the generalization capabilities of LLMs (Brown et al., 2020; Touvron et al., 2023a;b) to empower agent systems with tools (Qin et al., 2023b; Schick et al., 2024; Shen et al., 2024) is an essential task (Xi et al., 2023; Wang et al., 2023b). Schick et al. (2024) design a framework to allow LLM to use external APIs to complete tasks. Qin et al. (2023b) build related benchmarks to evaluate performance in such tasks. In tasks involving interaction with the environment, Shinn et al. (2024) enhance the agent's ability through language feedback signals. Wei et al. (2022b) employ Chain-of-Thought prompting method to optimize the reasoning capabilities of the LLM agent. However, the hallucination and interpretability challenges (Zhang et al., 2023) of LLMs also accompany these systems. In this work, we prove that the perspective of causal architecture can reduce the reliance on priors and enhance the robustness of inferences.

## 6 CONCLUSION

In this work, we introduce ADAM, an embodied causal agent in open-world environments. ADAM innovatively incorporates CD with embodied exploration, significantly improving the accuracy of CD while enhancing the efficiency and interpretability of embodied exploration. Without relying on prior knowledge, ADAM demonstrates strong robustness, and its multimodal perception closely aligns with human approaches. Our work sets a foundation for developing autonomous agents that can understand and manipulate environments in a causal manner.

ACKNOWLEDGEMENT

We thank all the anonymous reviewers and area chair for their valuable feedback throughout the review process. We also appreciate Zhuangyan Zhang for his contribution to some of the artistic elements in Fig. 2. This work is supported by the Shanghai Artificial Intelligence Laboratory.

REPRODUCIBILITY STATEMENT

We provide detailed descriptions of our experimental setup in Section 4.1, and we use the same OpenAI API (gpt-4-0125-preview) for all LLM-based methods to ensure comparability and reproducibility. We have open-sourced the code of ADAM at `https://github.com/OpenCausaLab/ADAM`.

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

## A   LLMs' PRIOR KNOWLEDGE ON MINECRAFT

We utilize the Minecraft crafting recipes to construct an MC-QA dataset (introduced in Appendix B), aiming to evaluate the prior knowledge of various LLMs on Minecraft. We test and determine that 0.3 is the optimal temperature as shown in Fig. 9a. Then we test various LLMs on this dataset. The GPT series (Ouyang et al., 2022; Achiam et al., 2023) show significantly stronger Minecraft prior knowledge than other LLMs as shown in Fig. 9b.

Utilizing this dataset to fine-tune LLMs can improve their prior knowledge on Minecraft as shown in Fig. 9c. On the other hand, by modifying the crafting recipes in Minecraft, we can make the SOTA LLMs (*e.g.*, GPT series) have no prior of this modified environment. This setup enables us to distinctively analyze the roles of *prior knowledge* and *inference capability* as shown in Fig. 9d, which serves as the basis of our ablation study.

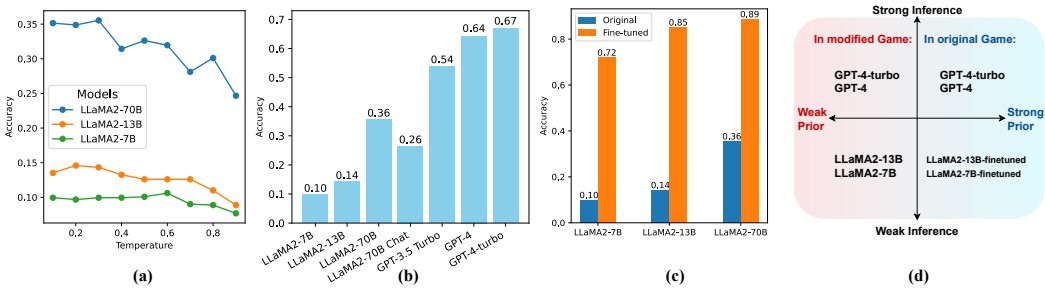

Figure 9: (a) LLaMA2 (Touvron et al., 2023b) demonstrates optimal accuracy in answering crafting recipes at a temperature of 0.3, measured as the ratio of correct answers to total questions. (b) Performance of open-source LLMs and GPT series models, showcasing their inherent prior knowledge of Minecraft. (c) Illustration of the improvement in performance for open-source LLMs fine-tuned with the MC-QA dataset. (d) Categorizing the LLMs into four types based on their prior knowledge of Minecraft recipes and inference capabilities.

## B   MC-QA DATASET

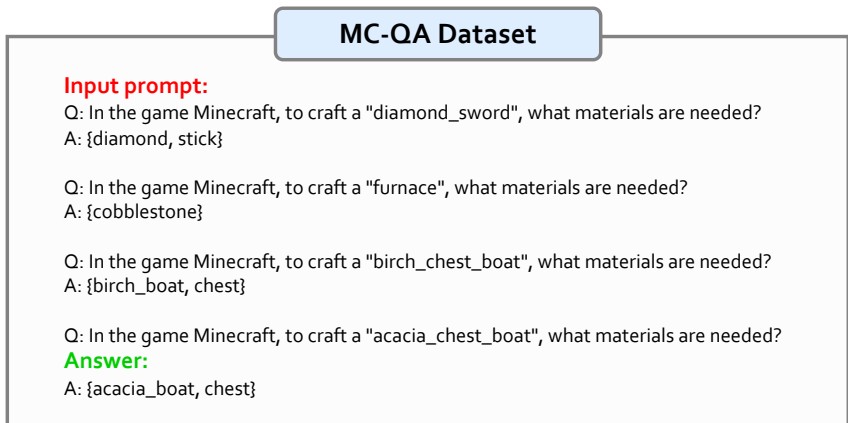

Figure 10: An example of the questions in MC-QA dataset.

Minecraft is an open-world game with a high degree of freedom. Players can freely gather materials (*e.g.*, log 🪵), mine ore (*e.g.*, `iron_ore` ⬛), craft tools (*e.g.*, `wooden_pickaxe` ⛏), and more. Crafting recipes (*e.g.*, `planks` 🟫 + `sticks` ∕ → `wood_pickaxe` ⛏) are the main knowledge in the Minecraft game, and serve as the basis for players to climb the technology tree and obtain more advanced items in the game.

Given this, LLMs' mastery of crafting recipes can reflect the strength of their prior knowledge in Minecraft. We utilize the crafting recipes in Minecraft (version 1.19) to create the MC-QA dataset. An example of the QA pairs in the dataset is shown in Fig. 10. The questions in this dataset ask for the crafting ingredients required to obtain higher-level items in the technology tree, and the answers are the ingredient items. LLMs need to provide their answers in the specified format. The order of the items in the answer is not required. For each question, we provide 3 examples to help LLMs understand the QA task and the format of the answers. For situations where there are multiple ways to craft the same item, we take them all into account to avoid the model being biased toward a fixed understanding of the game. The dataset contains 754 QA pairs on the knowledge of obtaining items in Minecraft.

## C   AGENT IN MINECRAFT

RL explorations in Minecraft agents focus on the efficient use of data (Baker et al., 2022; Fan et al., 2022), hierarchical RL design (Lin et al., 2021; Mao et al., 2022), innovative architecture modeling (Hafner et al., 2023), *etc.*. Hafner et al. (2023) use world models to achieve a general and scalable RL without human data or curricula. Much work (Guss et al., 2019; 2021; Kanervisto et al., 2022; Fan et al., 2022) has made different simplifications for the Minecraft environment to facilitate the RL agent systems. The MineDojo framework (Fan et al., 2022) provides an internet-scale knowledge database and game environments for CLIP model (Radford et al., 2021) and RL training. These efforts provide efficient optimization for agent sampling and interaction, but there is still a gap when compared to commercial Minecraft games with complete game features like in **VOYAGER** (Wang et al., 2023a) and our work.

The reasoning capabilities and rich prior knowledge of LLMs have contributed to much work on Minecraft agents (Yuan et al., 2023; Zhu et al., 2023; Wang et al., 2023a; Qin et al., 2023a; Nottingham et al., 2023; Wang et al., 2023d;c). VOYAGER (Wang et al., 2023a) and **GITM** (Zhu et al., 2023) use LLMs' prior knowledge of Minecraft and environment feedback to complete exploration tasks in a text-based manner. Qin et al. (2023a) leverage MLLMs to introduce visual information as the contextual basis for action execution. These methods more or less rely on prior knowledge of Minecraft. Our ADAM shows effectiveness even when the game rules are modified.

There are also agents that operate in non-commercial Minecraft environments and utilize frame-level control, including **Plan4MC** (Yuan et al., 2023), **DEPS** (Wang et al., 2023d), **JARVIS-1** (Wang et al., 2023c), **OmniJARVIS** (Wang et al., 2024), and **Optimus-1** (Li et al., 2024). They need approximately $10^4$ steps for a task, in contrast to the action-level control in our system, which involves around $10^2$ steps for a task. Furthermore, these RL methods require training stages, whereas our system does not involve weight updates. Notably, JARVIS-1 incorporates crafting recipes **as an integral part** of the system, utilizing prior knowledge rather than learning from scratch. Given the lack of comparable open-source MLLM-based Minecraft agent, comprehensive ablation studies on ADAM itself — including the ablation of TM and SD (Section 3.4, Tab. 1), LLM-based CD (Tab. 4), and intervention-based CD (Fig. 8) — establish a robust baseline for MLLM agents in Minecraft.

Due to the complexity of the Minecraft environment, **Crafter** (Hafner, 2022) simplifies the game space while maintaining the game rules of Minecraft. Crafter is a 2D game environment with crafting routes similar to Minecraft. Agents such as **SPRING** (Wu et al., 2024) and **Mars** (Tang et al., 2024) have been developed on this environment.

In the original implementation of VOYAGER, the environment feedback provides **detailed information** such as crafting recipe errors (*e.g.*, *"I cannot make an iron chestplate because I need: 7 more iron ingots."*). We retain this informative feedback in all our experiments with VOYAGER, even in environments with modified crafting recipes where the feedback may not align with the changes.

Tab. 5 shows the environmental information used by ADAM and VOYAGER. This setup is used in our experiments at Sec. 4.3. VOYAGER does not have a visual input and requires omniscient metadata (Meta) which is not explicitly exposed to human players, while ADAM utilizes visual input and other observable information to make decisions.

| | **VOYAGER** | **VOYAGER w/o Meta** | **ADAM** | **ADAM w/o MLLM** |
|---|---|---|---|---|
| Observation Space | Environment Feedback, Inventory, Meta | Environment Feedback, Inventory | Pixels, Inventory | Inventory |
| Action Space | Code | Code | Discrete | Discrete |

Table 5: Comparison of environmental information used by ADAM and VOYAGER. VOYAGER does not have a visual input and needs omniscient metadata (Meta) which is not explicitly exposed to human players, while ADAM relies on the visual input and the information observable by human players to make decisions.

| **Action Type** | **Action-Item Dependency** | **Data Quality** | **Skill Level** |
|---|---|---|---|
| Collecting | Simple | Noise | Low |
| Crafting | Complex | Clean | High |
| Smelting | Complex | Noise | High |

Table 6: Three action types in our experiment setting. "Smelting" actions have complex causal subgraphs, often leading to omissions in LLM-based CD. "Collecting" actions have noisy sampling data, and the results of the LLM-based CD are often redundant. The "Crafting" actions have complex causal subgraphs and clean sampling data.

# D    IMPLEMENTATION DETAILS

## D.1    PROMPT

The prompt for LLM-based CD is shown in Fig. 11, which is composed of 5 components: (1) Role Playing, which assigns a specific role to the LLM; (2) Problem Setting, which provides specific details of the inference task; (3) Letter Mapping, which involves mapping item names to letters, a simplification that facilitates the formatted output; (4) Few-shot Prompting, which involves providing the LLM with several inference examples in chain-of-thought (Wei et al., 2022b) style; (5) Data $D_a$, which is the data collected by the interaction module for inference.

## D.2    ACTION SPACE AND MOVEMENT SPACE

We have implemented 41 discrete actions and 6 movements to ensure the agent can freely explore the Minecraft world through a diverse range of their combinations. The actions can be divided into three categories: "Smelting", "Collecting", and "Crafting". "Smelting" actions have complex causal subgraphs, often leading to omissions in LLM-based CD. "Collecting" actions have noisy sampling data, and the results of the LLM-based CD are often redundant. The "Crafting" actions have complex causal subgraphs and clean sampling data.

The movement space corresponds to the low-level movement control visible to human players, including moving forward / backward, lowering / raising the agent's coordinates and turning left / right.

# E    ROBUSTNESS

Tab. 7 shows our experiment results in the modified Minecraft environment where the crafting recipes are altered. ADAM can maintain its performance as it is equipped with CD methods, whereas agents that rely on prior knowledge struggle to explore efficiently. The result demonstrates the robustness and generalization capabilities of our ADAM architecture.

You are a Minecraft game analysis assistant. Our task is to infer the effect of an action and explore the causal relationship by analyzing the items consumed and generated before and after an action. An action is a single, atomic action consisting of
1. "Crafting" type actions, that is, combining raw materials into an item.
2. "Collecting" type actions, that is, collecting certain items. There may be some by-products in this process.
3. "Smelting" type actions, that is, consuming fuel and raw metal materials, and obtaining smelted items. A furnace is needed.

It's possible that the action didn't have the expected effect, or that some additional items were collected, and we'll give multiple sampling records to ensure robustness. Such records all correspond to the execution of the same action. Your answer will only refer to the alphabetical codes of these items.
The answer should be in the format {Cause; Effect}

(Comment: The following is an example of the Observed Item Space. "m" and "n" are for few-shot learning and are not included in the technology tree to avoid prior)
Items involved in the game include
'a': 'log'
'b': 'planks'
'c': 'crafting_table'
'd': 'stick'
'e': 'wooden_pickaxe'
'f': 'cobblestone'
'g': 'stone_pickaxe'
'h': 'raw_iron'
'i': 'furnace'
'j': 'iron_ingot'
'k': 'iron_pickaxe'
'l': 'diamond'
'm': 'redstone'
'n': 'redstone_torch'

log corresponds to a variety of log named xx_log
planks correspond to a variety of planks named xx_planks

1. Initial items: iron_pickaxe, crafting_table ; Consumed items: ; Added items: redstone, cobblestone, cobbled_deepslate
2. Initial items: iron_pickaxe, crafting_table ;Consumed items: ; Added items: cobblestone, redstone, andesite
3. Initial items: iron_pickaxe, crafting_table ;Consumed items: ; Added items: granite
4. Initial items: iron_pickaxe, crafting_table ;Consumed items: ; Added items: cobblestone, cobbled_deepslate
5. Initial items: iron_pickaxe, crafting_table ;Consumed items: ; Added items: redstone, cobblestone

Your inference:
Because there is no item consumption, this is a "Collecting" task. Redstone is a higher-level mineral and needs to be collected with an iron_pickaxe ('k'). The rest of the items are considered additional items obtained during collection and will not be considered. So my answer is:
Cause; Effect={k; m}

1. Initial items: stick, redstone, crafting_table ; Consumed items: stick, redstone; Added items: redstone_torch
2. Initial items: stick, redstone, crafting_table ; Consumed items: ; Added items:
3. Initial items: stick, redstone, crafting_table ; Consumed items: redstone, stick; Added items: redstone_torch
4. Initial items: stick, redstone, crafting_table ; Consumed items: stick, redstone; Added items: redstone_torch
5. Initial items: stick, redstone, crafting_table ; Consumed items: redstone, stick; Added items: redstone_torch

Your inference:
Because redstone and sticks are being used to craft redstone torches, this is a "Crafting" task. In addition to raw materials, the "Crafting" task also requires crafting_table ('c'). So my answer is:
Cause; Effect={c, d, m; n}

(Our Data $D$)
Your inference:

Figure 11: The prompt for LLM-based CD. The contents in red will be replaced in the inference process.

# F  GENERALIZATION

The ADAM architecture is a general framework for embodied agents operating in various open-world environments including Minecraft. When adapting ADAM to other application scenarios, some modifications may be necessary:

(1) The world knowledge in Minecraft is the dependence between items and actions. Consequently, in this paper, items and actions are designed as causal graph nodes. When migrating to other environments, key elements related to the agent's task objectives can be similarly designed as causal graph nodes.

| Framework | Wooden Tool | Stone Tool | Iron Tool | Diamond |
|---|---|---|---|---|
| React w/ TM w/ SD | $91 \pm 34(2/3)$ | $139(1/3)$ | N/A $(0/3)$ | N/A $(0/3)$ |
| Reflexion w/ TM w/ SD | $76 \pm 28(2/3)$ | $120 \pm 40(2/3)$ | N/A $(0/3)$ | N/A $(0/3)$ |
| AutoGPT w/ TM w/ SD | $82 \pm 25(2/3)$ | $124(1/3)$ | N/A $(0/3)$ | N/A $(0/3)$ |
| VOYAGER | $95 \pm 33(2/3)$ | $152 \pm 43(2/3)$ | N/A $(0/3)$ | N/A $(0/3)$ |
| VOYAGER Guided | $108 \pm 35(2/3)$ | $176(1/3)$ | N/A $(0/3)$ | N/A $(0/3)$ |
| ADAM | $28 \pm 4(3/3)$ | $52 \pm 14(3/3)$ | $94 \pm 27(3/3)$ | $109 \pm 34(2/3)$ |
| ADAM Parallel | $\mathbf{15 \pm 2(3/3)}$ | $\mathbf{31 \pm 7(3/3)}$ | $\mathbf{54 \pm 14(3/3)}$ | $\mathbf{61 \pm 18(2/3)}$ |

Table 7: Performance in the modified Minecraft game. Each method has three trials for a maximum length of 200 steps. The success rate is depicted in the parentheses. ADAM can maintain its performance as it is equipped with CD methods, whereas agents that rely on prior knowledge struggle to explore efficiently. The result demonstrates the robustness and generalization capabilities of our ADAM architecture.

(2) In ADAM, the perception module utilizes a vision-based MLLM and does not rely on omniscient metadata. This allows the module to adapt well to other visual tasks. If conditions permit (*e.g.*, a robot equipped with LiDAR), the perception module can provide more precise information, potentially further improving performance.

(3) To model actions as a finite set of causal graph nodes, it is necessary to discretize the continuous action space. The granularity of this discretization should be determined based on the specific environment.

