# OpenReview forum: "ADAM: An Embodied Causal Agent in Open-World Environments"
_ICLR.cc/2025/Conference — ICLR 2025 Poster_

### Official Review · Reviewer_yx6R · 2024-10-29

**Soundness:** 1
**Presentation:** 3
**Contribution:** 2
**Rating:** 6
**Confidence:** 3

**Summary:**

The paper proposes a Minecraft agent called Adam that relies on a combination of (M)LLM inference and causal discovery. At the core of the method is a causal graph that represents the agent's expertise on the environment logic. The causal graph is proposed by an LLM and each relation in the graph is confirmed/disproved by environment interactions. The proposed method obtains diamonds faster and more reliably than prior methods. Moreover,

**Strengths:**

- the paper is well structured and clearly written
- the combination of LLM-prompting with CD seems to be quite novel
- the intervention-based refinement of the causal graph seems meaningful and practical

**Weaknesses:**

- unsupported claims: the authors claim their method has "excellent interpretability" and that their agent "closely aligns with human gameplay;" yet, I cannot find any empirical evidence supporting these claims.
- a runtime/memory analysis (be it theoretical or empirical) is completely missing, i.e., it is not clear at which cost the claimed SOTA results come
- the results presented in figure 1 are based on a modified causal graph claiming that this removes prior knowledge from LLMs; yet, it is unclear in how far this claim is true; it would be interesting to see an analysis akin to appendix A for the modified environment
- the choice of the acronym Adam is at best unfortunate as it coincides with one of the most influential machine learning papers (https://arxiv.org/abs/1412.6980) and could be mistaken for an attempt to tap unwitting citations as both paper titles start with "Adam: ..."
- insufficient reproducibility due to missing source code

**Questions:**

- ad interpretability claim: how does the interpretability of the presented method relate to the interpretability of baseline methods, e.g., voyager? isn't the interpretability of the method compromised by the lack of interpretability of the used LLMs?
- the proposed method is quite complex as it consists of 4 modules consisting of several submodules each and the presented ablation studies seem insufficient to rigorously justify such a complicated system; so how was the method designed? how much inspiration was drawn from prior work?
- Adam is claimed to be a "generalizable framework"; why not back this claim with a complementary application in another environment?
- what are possible limitations of the method? (the paper does not mention any)
- what do the error bars in tables 1, 2, 3 mean?

---

> ### Author Response · Authors · 2024-11-23
> **Responses to Reviewer yx6R (1/3)**
>
> ## Responses to Reviewer yx6R
>
> Thank you for your thoughtful review and constructive suggestions. We are delighted for your recognition of the novelty of our method and the clarity of our writing. We provide detailed replies to your comments and hope we can resolve your major concerns.
>
> **Concerns on Weaknesses:**
>
> > unsupported claims: their agent "closely aligns with human gameplay;"
>
> 1. We believe there may be a misunderstanding. **Our point is that ADAM's observation space 'closely aligns with human gameplay' without the use of metadata**, not that ADAM's gameplay level matches that of human players. Please refer to **Line 103**: "_We tackle the limitations of existing embodied agents. Our ADAM achieves excellent generalization capability_ _**without relying on prior knowledge or omniscient metadata like other LLM-based agents**, while_ _**exhibits human-like exploration**_ _as a general framework._" Unlike methods such as VOYAGER [1], which use metadata inaccessible to human players to guide decision-making, **ADAM employs a visual approach, making its process more similar to that of human players.** We have modified the relevant descriptions in the updated version to clarify it.
>
>
> > unsupported claims: "excellent interpretability"
>
> 2. ADAM's interpretability is based on its explicit representation of knowledge and decision-making through a causal graph. This approach differs fundamentally from the inherent knowledge of LLMs or RL agents, which often lack transparent and interpretable representations.
>
>      \
>      The causal graph enables us to **evaluate potential errors or biases in the agent's learned knowledge and provides a clear visualization of the decision-making process**. This systematic modeling of world knowledge is a novel contribution, as previous works typically rely on **learned neural network weights or rule-based knowledge bases**. By explicitly representing the causal relationships between world entities, our approach facilitates a more transparent and interpretable understanding of the agent.
>
>
> > a runtime/memory analysis (be it theoretical or empirical) is completely missing,
>
> 3. I understand your concern. All of our comparisons use **the same LLM API** (GPT-4-turbo), and **computations are** **performed externally rather than** **locally**. For the overhead of a single step, all our performance experiments are conducted on Mineflayer [2], ensuring that **the definition and calculation metrics of steps are** **consistent across all** **baselines**, including the number of API calls. We follow the evaluation metrics consistent with VOYAGER and other Minecraft agents.
>
>
> > insufficient reproducibility due to missing source code
>
> 4. We provide our source code on an anonymous website to meet the requirements of double-blind review. https://anonymous.4open.science/r/ADAM_anonymous2-B0C3

---

> ### Author Response · Authors · 2024-11-23
> **Responses to Reviewer yx6R (2/3)**
>
> **Questions:**
>
> > How does the interpretability of the presented method relate to the interpretability of baseline methods, e.g., voyager?
>
> 1. ADAM's interpretability is rooted in its decision-making process, which relies on a causal graph, as previously discussed. Methods like VOYAGER [1] do not emphasize transparency or interpretability as strengths. Their interpretability is inherently limited because their knowledge relies heavily on the static prior knowledge of LLMs, rather than being acquired through interactive exploration and intervention-based causal discovery.
>
>
> > Isn't the interpretability of the method compromised by the lack of interpretability of the used LLMs?
>
> 2. ADAM's interpretability is primarily facilitated by its causal graph. The causal graph does not rely on the **static prior knowledge embedded in the LLM**, but rather is acquired through empirical knowledge gathered from **continuous interactions with the environment, reasoning, and experimentation**. The usage of LLMs in ADAM does not affect the correctness and interpretability of the causal graph obtained through our causal algorithm. Therefore, "the lack of interpretability of the used LLMs" will **not** affect ADAM's interpretability, which **remains independent of LLMs**. In contrast, methods that rely solely on LLMs are affected by their lack of transparency.
>
>
> > The proposed method is quite complex. The presented ablation studies seem insufficient to rigorously justify such a complicated system; so how was the method designed?
>
> 3. Each module in our proposed system serves a distinct function, forming a closed loop as shown in **Figure 2(Line 122)**. The **Interaction Module** enables the agent to execute actions from the action space and processes the agent's observable information into formatted records. The **Causal Model Module** constructs the causal graph, the **Controller Module** selects actions and completes tasks in the environment, and the **Perception Module** provides visual observation description.
>
>        Our ablation experiments include:
>
>     1. The ablation of the Perception Module (**Table 3, line 448**)
>
>     2. The ablation of the LLM-based CD (**Table 4, Line 493**) and intervention-based CD (**Figure 8,** **Line 499**) in Causal Model Module.
>
>     3. (**Table 1, Line 384**) The ablation of on the role of the **TM (Temporal Modeling) and SD (Subgraph Decomposition)** **(Line 311)** used in the Causal Model Module.
>
>
>       As for the other two modules, they are the backbone of embodied exploration. Ablating these modules would render the agent incapable of interacting with the environment, thereby precluding any discussion of performance.
>
>
> > How much inspiration was drawn from prior work?
>
> 4. We indeed drew inspiration from previous works, mainly in the development of the **Controller module**. The collaboration among Planner, Actor, and Memory is a well-established design in embodied agents, as outlined in the survey [3], where Brain (planning ability and memory), Perception, and Action form the general workflow of these agents. This design corresponds to the structure of our Controller module. Our novel contribution lies in enhancing this design to better accommodate the construction and utilization of causal graphs, as demonstrated in our experiments (**Figure 1, Line 68**). We hope that the clarifications provided above, the explanation of our module design's rationale, and the supplementary code materials will lead to a re-evaluation of our paper's soundness. Please feel free to ask if you have any further questions.
>
>
> > Adam is claimed to be a "generalizable framework"; why not back this claim with a complementary application in another environment?
>
> 5. ADAM can be generalized, and we have provided a detailed process for generalizing our approach to other domains in "**Appendix F, GENERALIZATION (L 961)**". Our initial intention was to use Minecraft as one of the most representative and challenging environments, with the detailed explanation in this scenario serving as a typical example for broader transfer. We have revised the relevant statement in the updated version for clarity.

---

> ### Author Response · Authors · 2024-11-23
> **Responses to Reviewer yx6R (3/3)**
>
> > What are possible limitations of the method? (the paper does not mention any)
>
> 6. Open-source MLLMs are currently insufficiently accurate to provide detailed information for complex tasks, **as discussed in my response 7 to Reviewer GNQs**. Existing MLLMs primarily assist in observing peripheral environments, which has been validated through our ablation experiments. However, human feedback remains necessary for Minecraft tasks requiring high accuracy, as also noted by VOYAGER [1]. To our knowledge, relying solely on vision (without metadata) for complex tasks like building remains unfeasible, and we are committed to advancing in this direction, including modeling causal relationships in such tasks. ADAM's decision-making process relies on its **memory, the causal graph it constructs, and the reasoning ability of LLMs,** with MLLM playing only a supporting role.
>
>
> > What do the error bars in tables 1, 2, 3 mean?
>
> 7. The error bars in tables 1, 2, and 3 represent the standard deviation of the steps over multiple runs of the experiment. They provide a measure of the variability of the results and allow for a more accurate comparison of the performance of our method with that of the baseline methods.
>
>
> We hope the above response addresses your concerns. If you find our revisions and responses helpful, we would greatly appreciate your consideration in raising the score to support our work. Please let us know if you have any further questions.
>
>
> [1] Wang, G., Xie, Y., Jiang, Y., Mandlekar, A., Xiao, C., Zhu, Y., ... & Anandkumar, A. (2023). Voyager: An open-ended embodied agent with large language models. _arXiv preprint arXiv:2305.16291_.
>
> [2] PrismarineJS. Prismarinejs/mineflayer, 2023a. URL https://github.com/PrismarineJS/mineflayer. https://github.com/PrismarineJS/mineflayer .
>
> [3] Xi, Z., Chen, W., Guo, X., He, W., Ding, Y., Hong, B., ... & Gui, T. (2023). The rise and potential of large language model based agents: A survey. _arXiv preprint arXiv:2309.07864_.

---

> > ### Comment · Reviewer_yx6R · 2024-11-25
> >
> > Thank you for your answer. I especially appreciate the link to the code repo and adjusted my score.

---

> > > ### Author Response · Authors · 2024-11-25
> > > **Thank you for your affirmation**
> > >
> > > Thank you for your affirmation! We will continue to contribute to the open source community. If you have any other questions, please feel free to ask.

---

### Official Review · Reviewer_b8Jp · 2024-11-03

**Soundness:** 3
**Presentation:** 3
**Contribution:** 3
**Rating:** 6
**Confidence:** 3

**Summary:**

This paper introduces ADAM (An emboDied causal Agent in Minecraft) - an agent architecture that autonomously explores, learns causal world knowledge, and executes complex tasks in Minecraft from multimodal inputs. The system consists of four main components: interaction module, causal model module, controller module, and perception module. The interaction module samples actions and records observations. The causal model module infers causal relationships and constructs causal subgraphs for each action.
The key innovation is integrating causal discovery methods with embodied exploration, enabling the agent to learn accurate causal relations from scratch without relying on prior knowledge.

**Strengths:**

1. The incorporation of causal discovery methods in a modular framework is a novel in LLM-based embodied exploration, and it does not rely on privileged information unlike prior work
2. The paper demonstrate strong empirical results with well-designed experiments, that led to significantly faster discovery of skills in Minecraft. Performance in modified environments where prior knowledge is invalid did not degrade performance too much demonstrates causal learning is indeed effective. Method also does not require meta-data.
3. The experiments are solid with multiple baselines and includes comprehensive ablation studies as well as detailed analysis of failure cases

**Weaknesses:**

1. One concern is whether ADAM scales with more complex world and causal graph for intervention-based causal discovery (CD).
2. Interestingly the paper proposes a multimodal agentic framework but all the baselines compared to are text-based frameworks. It would be good to have at least one multi-modal baseline, e.g. [1] as this is also cited by the authors.

[1] Wang, Z., Cai, S., Liu, A., Jin, Y., Hou, J., Zhang, B., ... & Liang, Y. (2023). Jarvis-1: Open-world multi-task agents with memory-augmented multimodal language models. arXiv preprint arXiv:2311.05997.

**Questions:**

See previous

---

> ### Author Response · Authors · 2024-11-23
> **Responses to Reviewer b8Jp**
>
> ## Responses to Reviewer b8Jp
>
> Thank you for your thoughtful review and recognition of our contributions. We are glad that you found our incorporation of causal discovery methods in a modular framework to be novel and effective. We would like to address your concerns and questions.
>
> **Concerns on Weaknesses:**
>
> > One concern is whether ADAM scales with more complex world and causal graph for intervention-based causal discovery (CD).
>
> 1. This is possible. We have provided a detailed process for generalizing our approach to other domains in "**Appendix F, GENERALIZATION (L 961)**". We briefly outline the steps here:
> 	1. Identify key elements related to task objectives and represent them as causal graph nodes.
> 	2. If additional observational data is available (e.g., LiDAR), the perception module can potentially be enhanced beyond simply relying on visual input.
> 	3. Discretize the continuous action space into a finite set of causal graph nodes, with granularity tailored to the specific environment.
>
>
> > It would be good to have at least one multi-modal baseline, e.g. Jarvis-1 [1] as this is also cited by the authors.
>
> 2. Most current MLLM agent works explicitly utilize prior knowledge, and thus there is no baseline that can be fairly compared to ADAM. For instance, Jarvis-1 [1], as referenced, **explicitly uses recipe knowledge** (the core tech tree) as part of its agent system (see https://github.com/CraftJarvis/JARVIS-1/tree/main/jarvis/assets/recipes). **This is exactly what ADAM does not rely on****, as it learns entirely** **from scratch.** Without this predefined knowledge, Jarvis-1 would fail to function completely, making a direct comparison with ADAM infeasible. This further underscores the superiority of ADAM.
>
>
> We hope the above response addresses your concerns. If you find our revisions and responses helpful, we would greatly appreciate your consideration in raising the score to support our work. Please let us know if you have any further questions.
>
>
>
> [1] Wang, Z., Cai, S., Liu, A., Jin, Y., Hou, J., Zhang, B., ... & Liang, Y. (2023). Jarvis-1: Open-world multi-task agents with memory-augmented multimodal language models. _arXiv preprint arXiv:2311.05997_.

---

### Official Review · Reviewer_Gehh · 2024-11-03

**Soundness:** 2
**Presentation:** 1
**Contribution:** 2
**Rating:** 5
**Confidence:** 3

**Summary:**

The paper introduces an architecture for agents that play the game of Minecraft, based on combining Large Language Models with causal inference. Featuring different modules (e.g., planner, actor, perception), the method is based on inferring causal graphs related to the various crafting dependencies of Minecraft's technology tree, and on enabling an agent to use the knowledge about these dependencies for progressing in the game. The method is evaluated against similar LLM-based methods (albeit using a different action space).

**Strengths:**

- The method seems to be the first approach that combines casual inference with LLM agents in code-based action spaces, which is a potentially very important direction for future research.
- The method performs quite well for inferring causal graphs on Minecraft, and it seems to provide a way for agents to take advantage of those causal graphs.

**Weaknesses:**

- The presentation of the method is quite high-level and does not help the reader understand how the method actually works in practice. When the different "modules" are introduced, it is not clear a priori what they actually are. Are they just prompts and specifications to a GPT4 model? If so, it could be beneficial to show one of the prompts earlier, to guide the understanding of the rest of the paper.
- The comparisons in the paper are unclear, due to the choice of a specific action space that is different from the one used in previous work. Indeed, while the paper mostly discusses the "observation space" difference compared to the setting usually employed in reinforcement learning papers, one crucial difference is the one in action space. For instance, Voyager works in an extremely more high-level action space compared to DreamerV3. This is not accurately depicted in the current version of the paper.
- It could be surprising to observe that an off-the-shelf open model is accurately describe an observation to the level of providing enough information for an actor to take the optimal action, especially in an environment that is as visually rich as Minecraft. An ad-hoc evaluation of this specific capability would strengthen the paper.
- The method seems to be highly Minecraft-specific, and the paper does not extensively discuss how it could be generalized to other domains.

**Questions:**

- Would the method generalize to other environments? If so, what are the assumptions and requirements for the application of the method to a new environment?
- How does the method compare to approaches trained with reinforcement learning? Is it possible to train an agent with reinforcement learning on the same action space that ADAM uses?
- What is the captioning performance of the perception module? What are its failure cases?

---

> ### Author Response · Authors · 2024-11-23
> **Responses to Reviewer Gehh (1/4)**
>
> ## Responses to Reviewer Gehh
>
> We are very grateful for your in-depth and thoughtful comments. We appreciate your recognition of the novelty and performance of our method. We are here to provide detailed replies to your comments and hope we can resolve your major concerns.
>
> **Concerns on Weaknesses:**
>
> > The presentation of the method is quite high-level and does not help the reader understand how the method actually works in practice. When the different "modules" are introduced, it is not clear a priori what they actually are.
>
> 1. We respectively disagree with your assessment. Our presentation is not "quite high-level", but rather detailed and thorough. We provided an in-depth introduction to each module's function, including specific examples to illustrate how they are working. Furthermore, we used detailed diagrams to explain how each module works and how they interact (**Interaction Module: Figure 3; Causal model module: Figure 4 and Figure 5; Controller module: Figure 6**). Notably, **all** **the other three reviewers** **have given positive evaluations** **regarding the clarity of** **our** **presentation**, and we believe it effectively conveys the intended details. If you have any further questions, please feel free to ask, and we will do our best to clarify.
>
>      \
>      Taking the **Interaction Module** as an example, **Figure 3** provides a step-by-step illustration of the process. Starting from the initial action space {gatherWoodLog}, the figure demonstrates how the action space is sampled and how the sampled data is subsequently processed into a record. Similar detailed examples are provided for other modules to ensure a comprehensive understanding.
>
>
> > Are they just prompts and specifications to a GPT4 model? If so, it could be beneficial to show one of the prompts earlier, to guide the understanding of the rest of the paper.
>
> 2. The modules of ADAM extend beyond prompt engineering; they are components **designed to interact with the environment in an embodied manner, as well as to extract, validate, and organize causal knowledge.** Fundamentally, they rely on logical execution and the mathematical representation of environmental knowledge, rather than merely using LLMs with designed prompts. Notably, modules such as the Interaction Module and the intervention-based causal discovery (CD) in the Causal Model Module do not utilize LLMs at all.
>
>      \
>      For example, the Interaction Module can determine how to sample the action space based on the task and the causal graph learned by other modules, and can parse the data obtained from the environment into formatted data for other modules. Detailed examples illustrating the operational logic of each module are provided (see above), with similar approaches applied in the other modules.
>
>
> > The comparisons in the paper are unclear, due to the choice of a specific action space that is different from the one used in previous work. Indeed, while the paper mostly discusses the "observation space" difference compared to the setting usually employed in reinforcement learning papers, one crucial difference is the one in action space.
>
> 3. Thank you for your suggestion. We provide a detailed description of the differences between various Minecraft agents in the appendix (**L 875, Appendix C AGENT IN MINECRAFT**).
>
>      \
>      Here, we elaborate on the action space of current Minecraft agents. The action space is primarily determined by the environment in which the agent is deployed. Currently, there are three main types of 3D Minecraft environments used in agent research: **MineRL [1], MineDOJO [2], and Mineflayer [3].**
>
>      1. **MineRL [1]** is represented by work such as **VPT [4], DreamerV3 [5], STEVE-1 [6], GROOT [7], Jarvis-1 [8], OmniJARVIS [9], and Optimus-1 [10]**. MineRL utilizes **low-level actions** based on keyboard and mouse operations, with some methods introducing high-level actions, such as Optimus-1 [10]'s addition of craft and smelt actions.
>
>     2. **MineDOJO [2]** is represented by methods such as **MineDOJO [2] (low-level action), DEPS [11] (discrete action code), and MP5 [12] (discrete action code)**.
>
>     3. **Mineflayer [3]**, on the other hand, **does not provide a Minecraft environment itself, but rather requires the combination of a commercial Minecraft**, thereby offering the most complex environment that has features identical to which human players experienced. Representative work using Mineflayer includes **VOYAGER [13]** **(JavaScript** **code)** and our **ADAM** **(discrete action code)**.

---

> ### Author Response · Authors · 2024-11-23
> **≡ Responses to Reviewer Gehh (2/4)**
>
> > It could be surprising to observe that an off-the-shelf open model is accurately describe an observation to the level of providing enough information for an actor to take the optimal action, especially in an environment that is as visually rich as Minecraft.
>
> 4. Open-source MLLMs are currently insufficiently accurate to provide detailed information for complex tasks, **as discussed in** **my response 7 to Reviewer GNQs**. Existing MLLMs primarily assist in observing peripheral environments, which has been validated through our ablation experiments. However, human feedback remains necessary for Minecraft tasks requiring high accuracy, as also noted by VOYAGER [13]. To our knowledge, relying solely on vision (without metadata) for complex tasks like building remains unfeasible, and we are committed to advancing in this direction, including modeling causal relationships in such tasks. ADAM's decision-making process relies on its **memory, the causal graph it** **constructs****, and the reasoning ability of LLMs****,** with MLLM playing only a supporting role.
>
>
> > The method seems to be highly Minecraft-specific, and the paper does not extensively discuss how it could be generalized to other domains.
>
> 5. We have provided a detailed process for generalizing our approach to other domains in "**Appendix F, GENERALIZATION (L 961)**". We briefly outline the steps here:
> 	1. Identify key elements related to task objectives and represent them as causal graph nodes.
> 	2. If additional observational data is available (e.g., LiDAR), the perception module can potentially be enhanced beyond simply relying on visual input.
> 	3. Discretize the continuous action space into a finite set of causal graph nodes, with granularity tailored to the specific environment.

---

> ### Author Response · Authors · 2024-11-23
> **Responses to Reviewer Gehh (3/4)**
>
> **Questions:**
>
> > Would the method generalize to other environments? If so, what are the assumptions and requirements for the application of the method to a new environment?
>
> 1. We have provided a detailed process for generalizing our approach to other domains in "**Appendix F, GENERALIZATION (L 961)**". See the above response.
>
>
> > How does the method compare to approaches trained with reinforcement learning?
>
> 2. Thank you for the good question. RL agents and LLM-based agents focus on addressing different challenges. RL is well-suited for learning the **execution of specific actions**, such as gathering wood or mining minerals, but struggles with long-term dependencies. SOTA RL systems often employ hierarchical designs (e.g., CDHRL [14]), which require substantial human involvement. Effective RL models typically require (often) millions of learning steps, and their knowledge is embedded within network weights, **making it difficult to express in a structured form (e.g., a graph)**. This black-box nature limits controllability, transparency, and interpretability. In contrast, LLM-based agents focus on addressing long-term problems, with some also integrating RL models as execution units (Plan4MC [15] , Jarvis-1 [8], etc.).
>
>      \
>      One of the major contributions of our work is to **alleviate the reliance on prior knowledge** **in** **LLM agents**. Current Minecraft agents use LLMs that have been **well-trained on the game's knowledge** to complete related tasks, which has introduced bias. To address this, ADAM explores Minecraft from scratch and stores knowledge in the form of causal graphs, which enhances robustness and interpretability.
>
>
> > Is it possible to train an agent with reinforcement learning on the same action space that ADAM uses?
>
> 3. Theoretically, training an RL agent in our environment is feasible. However, using our commercial Minecraft to support large-scale data sampling would be costly, as each step takes approximately 2 minutes. With an action space of 41 actions, **the permutations and combinations required for RL** **to find a successful decision path** **would necessitate significantly more steps, making it impractical.**
>
>
> > What is the captioning performance of the perception module? What are its failure cases?
>
> 4. See my response 4. MLLM serves only an auxiliary role and is **not well-suited for tasks requiring fine-grained positional accuracy** (e.g., building a house). While using metadata could improve this aspect, **our goal is to develop agents without relying on metadata**.
>
>
> We hope the above response addresses your concerns. If you find our revisions and responses helpful, we would greatly appreciate your consideration in raising the score to support our work. Please let us know if you have any further questions.

---

> ### Author Response · Authors · 2024-11-23
> **Responses to Reviewer Gehh (4/4)**
>
> [1] Guss, W. H., Houghton, B., Topin, N., Wang, P., Codel, C., Veloso, M., & Salakhutdinov, R. (2019). Minerl: A large-scale dataset of minecraft demonstrations. _arXiv preprint arXiv:1907.13440_.
>
> [2] Fan, L., Wang, G., Jiang, Y., Mandlekar, A., Yang, Y., Zhu, H., ... & Anandkumar, A. (2022). Minedojo: Building open-ended embodied agents with internet-scale knowledge. _Advances in Neural Information Processing Systems_, _35_, 18343-18362.Advances in Neural Information Processing Systems, 35:18343–18362, 2022.
>
> [3] PrismarineJS. Prismarinejs/mineflayer, 2023a. URL https://github.com/PrismarineJS/mineflayer.
>
> [4] Baker, B., Akkaya, I., Zhokov, P., Huizinga, J., Tang, J., Ecoffet, A., ... & Clune, J. (2022). Video pretraining (vpt): Learning to act by watching unlabeled online videos. _Advances in Neural Information Processing Systems_, _35_, 24639-24654.
>
> [5] Hafner, D., Pasukonis, J., Ba, J., & Lillicrap, T. (2023). Mastering diverse domains through world models. _arXiv preprint arXiv:2301.04104_.
>
> [6] Lifshitz, S., Paster, K., Chan, H., Ba, J., & McIlraith, S. (2024). Steve-1: A generative model for text-to-behavior in minecraft. _Advances in Neural Information Processing Systems_, _36_.
>
> [7] Cai, S., Zhang, B., Wang, Z., Ma, X., Liu, A., & Liang, Y. (2023). Groot: Learning to follow instructions by watching gameplay videos. _arXiv preprint arXiv:2310.08235_.
>
> [8] Wang, Z., Cai, S., Liu, A., Jin, Y., Hou, J., Zhang, B., ... & Liang, Y. (2023). Jarvis-1: Open-world multi-task agents with memory-augmented multimodal language models. _arXiv preprint arXiv:2311.05997_.
>
> [9] Wang, Z., Cai, S., Mu, Z., Lin, H., Zhang, C., Liu, X., ... & Liang, Y. (2024). OmniJARVIS: Unified Vision-Language-Action Tokenization Enables Open-World Instruction Following Agents. _arXiv preprint arXiv:2407.00114_.
>
> [10] Li, Z., Xie, Y., Shao, R., Chen, G., Jiang, D., & Nie, L. (2024). Optimus-1: Hybrid multimodal memory empowered agents excel in long-horizon tasks. _arXiv preprint arXiv:2408.03615_.
>
> [11] Wang, Z., Cai, S., Chen, G., Liu, A., Ma, X., & Liang, Y. (2023). Describe, explain, plan and select: Interactive planning with large language models enables open-world multi-task agents. _arXiv preprint arXiv:2302.01560_.
>
> [12] Qin, Y., Zhou, E., Liu, Q., Yin, Z., Sheng, L., Zhang, R., ... & Shao, J. (2024, June). Mp5: A multi-modal open-ended embodied system in minecraft via active perception. In _2024 IEEE/CVF Conference on_ _Computer Vision and Pattern Recognition_ _(__CVPR__)_ (pp. 16307-16316). IEEE.
>
> [13] Wang, G., Xie, Y., Jiang, Y., Mandlekar, A., Xiao, C., Zhu, Y., ... & Anandkumar, A. (2023). Voyager: An open-ended embodied agent with large language models. _arXiv preprint arXiv:2305.16291_.
>
> [14] Hu, X., Zhang, R., Tang, K., Guo, J., Yi, Q., Chen, R., ... & Chen, Y. (2022). Causality-driven hierarchical structure discovery for reinforcement learning. _Advances in Neural Information Processing Systems_, _35_, 20064-20076.
>
> [15] BAAI, P. (2023). Plan4mc: Skill reinforcement learning and planning for open-world minecraft tasks. _arXiv preprint arXiv:2303.16563_.

---

### Official Review · Reviewer_GNQs · 2024-11-04

**Soundness:** 2
**Presentation:** 3
**Contribution:** 2
**Rating:** 1
**Confidence:** 5

**Summary:**

This paper introduces ADAM, an autonomous agent for open-world environments like Minecraft that builds a causal graph from scratch to improve interpretability and performance without relying heavily on pretrained knowledge. Through a combination of interaction, causal reasoning, planning, and multimodal perception, ADAM outperforms existing agents in task success and adaptability, even in modified game settings. ADAM’s approach establishes a new standard for causal reasoning in embodied agents.

**Strengths:**

1.	The figures in the paper are well-done and enhance clarity, making the content easier to understand.
2.	The experiments conducted in Minecraft show a higher success rate than those achieved by Voyager.

**Weaknesses:**

1.	The paper appears to be hastily prepared, as it contains numerous typos and minor errors, such as inconsistencies between “Fig.” and “Figure” references and improper usage of quotation marks in Table 4’s caption. I recommend that the authors carefully review and correct these issues.
2.	The major issue lies in the extensive use of pretrained language models that already incorporate substantial knowledge of Minecraft. Since language models may internally form a comprehensive causal graph of the game world, primarily in linguistic form, the proposed additional causal graph construction might be redundant. I suggest that the authors explore scenarios with completely altered world rules in Minecraft to test the validity of models like GPT in such modified environments, perhaps using a setting like “Mars.” Alternatively, they could consider using a language model entirely devoid of Minecraft knowledge, though this may be challenging to achieve.
3.	The agent’s modular design is nearly identical to Voyager, with the primary addition being the causal graph. Experimentally, however, it does not show significant advantages over Voyager, as it does not complete tasks that Voyager was unable to.
4.	The causal graph generated by the model ADAM is quite similar to the hybrid knowledge graph in memory described in [1]. The authors should clarify the differences.
5.	Additionally, the current causal graph is entirely object-centric. In open-world Minecraft, there are many open-ended tasks, such as building and farming, which are not strictly object-centric. This limitation restricts ADAM’s generalization capability in open-ended tasks.
6.	Several relevant works are not cited, including:

	[1] Optimus-1: Hybrid Multimodal Memory Empowered Agents Excel in Long-Horizon Tasks

	[2] Mars: Situated Inductive Reasoning in an Open-World Environment, NeurIPS 2024

	[3] OmniJARVIS: Unified Vision-Language-Action Tokenization Enables Open-World Instruction Following Agents, NeurIPS 2024

**Questions:**

See the weakness.

According to the author's reply, they refuse to admit the unfair comparisons they made during the rebuttal stage. I even doubt whether the author carefully reviewed the responses from all reviewers. Therefore, I have decided to change the score to strong reject.

I strongly recommend that the author carefully review the reviewer's comments and provide a serious response.

---

> ### Author Response · Authors · 2024-11-23
> **Responses to Reviewer GNQs (1/3)**
>
> ## Responses to Reviewer GNQs
>
> Thank you for your thoughtful review and constructive suggestions. We are delighted by your recognition of our paper's clear presentation and ADAM’s performance. We provide detailed replies to your comments and hope we can resolve your major concerns.
>
>
>
> **Concerns on Weaknesses:**
>
> > It contains numerous typos and minor errors. I recommend that the authors carefully review and correct these issues.
>
> 1. Thank you for your suggestion. We believe the phrase "numerous typos" may overstate the issue, as we have not identified many. Nonetheless, we will continue to review our paper carefully. Should you notice any additional typos, we welcome your input for further improvement.
>
>
> > The major issue lies in **the extensive use of pretrained language models that already incorporate substantial knowledge of Minecraft**. Since language models may internally form a comprehensive causal graph of the game world, primarily in linguistic form, the proposed additional causal graph construction might be redundant.
>
> 2. We respectfully disagree and believe there may be a misunderstanding of our fundamental experiment. Your statement appears contrary to the findings presented in our paper. ADAM's robustness in the modified environment stems from its ability to construct causal graphs **without relying on prior knowledge** (**Fig. 1 (Line 68), Robustness (Line 428)**). Specifically, ADAM uses the **reasoning ability** of LLM **(Line 268)** and **intervention-based causal method (Line 280)** to obtain the causal graph. This has been clearly demonstrated in our experiments and was highlighted by **Reviewer b8Jp** as strengths 1 and 2.
>
>     \
>     The issue you raised is actually **a** **limitation of existing methods** (such as VOYAGER [1], Jarvis-1 [2], OmniJARVIS [3]), and represents **a** **fundamental** **difference** between our approach and others. In the modified environment, the game rules (causal graph) of Minecraft are altered, **making prior knowledge even harmful**. Therefore, in such a modified environment, "LLMs incorporated substantial knowledge of Minecraft" is actually **mismatched** with the environment, and only **constructing causal graphs from scratch** allows for successful task completion. This is one of our most important experimental results (**Fig. 1 (Line 68), Table 7 (Line 1055)**), demonstrating ADAM's superior performance -- being the only approach capable of mining diamonds.
>
>     \
>     The causal graph constructed by ADAM is **not** derived from the prior knowledge embedded in LLMs; instead, it emerges from **an** **iterative** **intervention** **and reasoning process** enabled by the Interaction Module and the Causal Model Module. **This distinction is crucial****: prior knowledge from LLMs is inherently limited to static information, whereas ADAM's approach allows for adaptability in dynamic environments.** The learned causal graph provides ADAM with enhanced **robustness** to environmental changes, as demonstrated in our experiments **(Robustness, Line 428).**
>
> > I suggest that the authors explore scenarios with completely altered world rules in Minecraft to test the validity of models like GPT in such modified environments, perhaps using a setting like “Mars.”
>
> 3. Our robustness experiment **(Line 428)** **has already altered the core dependencies of the game**, and in the ablation experiment, we verified that the core of task performance is the reasoning ability of LLM, not prior knowledge.
> \
> \
>     Mars [4] uses Crafter [5], which **simplifies** the environment while **retaining** Minecraft's prior knowledge. In other words, Mars [4] (Crafter [5]) does not "(fully) alter world rules".
>
>
> > The agent’s modular design is nearly identical to Voyager, with the primary addition being the causal graph.
>
> 4. We respectfully disagree. The three key components of VOYAGER [1] -- Automatic Curriculum, Iterative Prompting Mechanism, and Skill Library -- are **fundamentally** **different from our** **approach**. We are unclear about the basis for your conclusion, as all our innovative designs are tailored to the process of **discovering** **(Line 205, 268), verifying (Line 280), and utilizing (Line 316) causal graphs from scratch**, which VOYAGER does not address.
>
>     \
>     While VOYAGER manages existing knowledge and skills via a skill library, ADAM employs causal graphs, representing **a significant difference in knowledge discovery, storage, and utilization.** Incorporating causal structures into embodied intelligence requires considerable effort, which constitutes our original contribution, as acknowledged by **reviewers Gehh and b8Jp**.

---

> ### Author Response · Authors · 2024-11-23
> **Responses to Reviewer GNQs (2/3)**
>
> > Experimentally, however, it does not show significant advantages over Voyager, as it does not complete tasks that Voyager was unable to.
>
> 5. Our method has demonstrated significant advantages over VOYAGER in both the **original (2.2× speedup, Line 440)** and **modified (4.6× speedup Line 68)** environments. **More importantly****, ADAM successfully obtains diamonds in the Modified environment, a feat that VOYAGER is unable to accomplish (Figure 1, Line 68).** You may have misinterpreted our ablation experiments **(Line 449)** and performance experiments **(Line 440)**.
>
>
> > The causal graph generated by the model ADAM is quite similar to the hybrid knowledge graph in memory described in Optimus-1
>
> 6. We wish to clarify that the game rules (tech tree) of Minecraft **are publicly available through the Minecraft community and various wikis, and are not exclusive to academic research.** You can verify this through image search engines. This is similar for all Minecraft-related works, provided they do not manually alter the game rules. Analogously, most StarCraft agent research also relies on the standard tech tree [6][7], allowing agents to compete with human players under the same conditions.
>
>      \
>      **The key point is how the agent obtains this rule**, which brings us back to the discussion in (2). Since the Minecraft tech tree has appeared too many times on the internet, previous LLM agents directly memorized this graph, while ADAM discovers it **from scratch**. We tested the baselines' performance in a modified Minecraft environment and **found that their performance immediately drops significantly (Line 1055 Table 7)**. ADAM solves this problem by discovering causal relationships from scratch to adapt to any changing environment.
>
>
> > The current causal graph is entirely object-centric. In open-world Minecraft, there are many open-ended tasks, such as building and farming, which are not strictly object-centric.
>
> 7. This limitation exists in current methods when metadata is not used. Descriptions generated by MLLMs struggle to precisely indicate the relative positions of items, **posing challenges for tasks requiring fine-grained positional accuracy** (e.g., building a house). While using metadata could potentially improve this, **our goal is to develop agents without relying on metadata**, recognizing that such information may not be accessible in real-world scenarios. We hope that future advancements in MLLMs will enhance their ability to describe relative positions, an effort that will likely require contributions from the broader community.
>
>
> > Several relevant works are not cited, including:
> >
> > [1] Optimus-1: Hybrid Multimodal Memory Empowered Agents Excel in Long-Horizon Tasks
> >
> > [2] Mars: Situated Inductive Reasoning in an Open-World Environment, NeurIPS 2024
> >
> > [3] OmniJARVIS: Unified Vision-Language-Action Tokenization Enables Open-World Instruction Following Agents, NeurIPS 2024
>
> 8. Thank you for reminding us, and we have added references to these papers in the updated version.
>
>
> We hope the above response addresses your concerns. If you find our revisions and responses helpful, we would greatly appreciate your consideration in raising the score to support our work. Please let us know if you have any further questions.

---

> ### Author Response · Authors · 2024-11-23
> **Responses to Reviewer GNQs (3/3)**
>
> [1] Wang, G., Xie, Y., Jiang, Y., Mandlekar, A., Xiao, C., Zhu, Y., ... & Anandkumar, A. (2023). Voyager: An open-ended embodied agent with large language models. _arXiv preprint arXiv:2305.16291_.
>
> [2] Wang, Z., Cai, S., Liu, A., Jin, Y., Hou, J., Zhang, B., ... & Liang, Y. (2023). Jarvis-1: Open-world multi-task agents with memory-augmented multimodal language models. _arXiv preprint arXiv:2311.05997_.
>
> [3] Wang, Z., Cai, S., Mu, Z., Lin, H., Zhang, C., Liu, X., ... & Liang, Y. (2024). OmniJARVIS: Unified Vision-Language-Action Tokenization Enables Open-World Instruction Following Agents. _arXiv preprint arXiv:2407.00114_.
>
> [4] Tang, X., Li, J., Liang, Y., Zhu, S. C., Zhang, M., & Zheng, Z. (2024). Mars: Situated Inductive Reasoning in an Open-World Environment. _arXiv preprint arXiv:2410.08126_.
>
> [5] Hafner, D. (2021). Benchmarking the spectrum of agent capabilities. _arXiv preprint arXiv:2109.06780_.
>
> [6] Vinyals, O., Ewalds, T., Bartunov, S., Georgiev, P., Vezhnevets, A. S., Yeo, M., ... & Tsing, R. (2017). Starcraft ii: A new challenge for reinforcement learning. _arXiv preprint arXiv:1708.04782_.
>
> [7] Samvelyan, M., Rashid, T., De Witt, C. S., Farquhar, G., Nardelli, N., Rudner, T. G., ... & Whiteson, S. (2019). The starcraft multi-agent challenge. _arXiv preprint arXiv:1902.04043_.

---

### Author Response · Authors · 2024-11-23
**Responses to All (1/2)**

## Responses to All
We thank all reviewers for their insightful comments and acknowledgment of our contributions. Reviewers have appreciated ADAM's **novel combination of causal inference with LLM agents** in embodied exploration (**Reviewer Gehh, b8Jp, yx6R**), and its ability to learn accurate causal relations **without prior knowledge or meta-data** (**Reviewer b8Jp**), while demonstrating better **performance** (**Reviewer GNQs, Gehh, b8Jp**). Reviewers also praised the paper's "**well structured and clearly written**" presentation (**Reviewer GNQs, yx6R**) and the "**strong empirical results**" with well-designed experiments which are "**solid with multiple baselines** and includes **comprehensive ablation studies**" (**Reviewer b8Jp**). **We have updated our paper** with the changes marked in red. We provide our source code on an anonymous website to meet the requirements of double-blind review: https://anonymous.4open.science/r/ADAM_anonymous2-B0C3

## Common Concerns

### 1. Experiment Setting

ADAM is deployed in a **commercial Minecraft environment** that has features identical to those experienced by human players, making it a much more challenging environment compared to **Minecraft-style 2D environments** (e.g., the Crafter [1] used in Mars [2], SPRING [3]). These 2D environments (a 64 × 64 discrete world) [1] has been greatly simplified compared to our **infinite 3D-world**.

Methods using **non-commercial** Minecraft environments (e.g., **MineRL [10] and MineDOJO [11]** used in DreamerV3 [4], Plan4MC [5], DEPS [6], JARVIS-1 [7] and OmniJARVIS [8]) generally require a large number of learning/exploration steps due to their **frame-level** **learning approach.** As such, **they mostly use success rates as their metrics rather than efficiency** **(i.e., steps),** **whereas our evaluation considers both success and efficiency**. Our experimental design tries to include all potentially comparable work. Please refer to "**Appendix C, AGENT IN MINECRAFT(Line 875)**" for details.




|            |                                    |                  |                                                            |                   |
| ---------- | ---------------------------------- | ---------------- | ---------------------------------------------------------- | ----------------- |
| **Method** | **Steps to Get Diamond**           | **Success Rate** | **Action Space**                                           | **Need Metadata** |
| JARVIS-1   | up to 36,000 (start from iron_axe) | 0.092            | low-level discrete + Equip + Craft + Smelt                 | Need              |
| OmniJARVIS | up to 12,000                       | 0.08 ± 0.04      | -                                                          | -                 |
| VOYAGER    | 75 ± 20 (up to 200)                | 2/3              | code                                                       | Need              |
| ADAM       | 34 ± 7                             | 3/3              | high-level discrete, whose names and effects are not known | Not need          |


### 2. Scalability with Other Environments

We have provided a detailed analysis in "**Appendix F, GENERALIZATION (L 961)**". We briefly outline the steps here:
1. Identify key elements related to task objectives and represent them as causal graph nodes.
2. If additional observational data is available (e.g., LiDAR), the perception module can potentially be enhanced beyond simply relying on visual input.
3. Discretize the continuous action space into a finite set of causal graph nodes, with granularity tailored to the specific environment.

### 3. Where does the causal graph come from?

The causal graph constructed by ADAM is **not** from the prior knowledge embedded in LLMs, but rather an **emergent product of the iterative sampling and reasoning process** mainly facilitated by the Interaction Module and the Causal Model Module. This distinction is crucial, as the former is inherently limited to the static knowledge encoded in the LLM, whereas the latter enables adaptability in dynamic environments. The learned causal graph, in turn, endows ADAM with enhanced **robustness** in the face of environmental changes as demonstrated in our experiments **(Robustness, Line 428).**

---

> ### Author Response · Authors · 2024-11-23
> **Responses to All (2/2)**
>
> [1] Hafner, D. (2021). Benchmarking the spectrum of agent capabilities. _arXiv preprint arXiv:2109.06780_.
>
> [2] Tang, X., Li, J., Liang, Y., Zhu, S. C., Zhang, M., & Zheng, Z. (2024). Mars: Situated Inductive Reasoning in an Open-World Environment. _arXiv preprint arXiv:2410.08126_.
>
> [3] Wu, Y., Min, S. Y., Prabhumoye, S., Bisk, Y., Salakhutdinov, R. R., Azaria, A., ... & Li, Y. (2024). Spring: Studying papers and reasoning to play games. _Advances in Neural Information Processing Systems_, _36_.
>
> [4] Hafner, D., Pasukonis, J., Ba, J., & Lillicrap, T. (2023). Mastering diverse domains through world models. _arXiv preprint arXiv:2301.04104_.
>
> [5] BAAI, P. (2023). Plan4mc: Skill reinforcement learning and planning for open-world minecraft tasks. _arXiv preprint arXiv:2303.16563_.
>
> [6] Wang, Z., Cai, S., Chen, G., Liu, A., Ma, X. S., & Liang, Y. (2024). Describe, explain, plan and select: interactive planning with LLMs enables open-world multi-task agents. _Advances in Neural Information Processing Systems_, _36_.
>
> [7] Wang, Z., Cai, S., Liu, A., Jin, Y., Hou, J., Zhang, B., ... & Liang, Y. (2023). Jarvis-1: Open-world multi-task agents with memory-augmented multimodal language models. _arXiv preprint arXiv:2311.05997_.
>
> [8] Wang, Z., Cai, S., Mu, Z., Lin, H., Zhang, C., Liu, X., ... & Liang, Y. (2024). OmniJARVIS: Unified Vision-Language-Action Tokenization Enables Open-World Instruction Following Agents. _arXiv preprint arXiv:2407.00114_.
>
> [9] Wang, G., Xie, Y., Jiang, Y., Mandlekar, A., Xiao, C., Zhu, Y., ... & Anandkumar, A. (2023). Voyager: An open-ended embodied agent with large language models. _arXiv preprint arXiv:2305.16291_.
>
> [10] Guss, W. H., Houghton, B., Topin, N., Wang, P., Codel, C., Veloso, M., & Salakhutdinov, R. (2019). Minerl: A large-scale dataset of minecraft demonstrations. _arXiv preprint arXiv:1907.13440_.
>
> [11] Fan, L., Wang, G., Jiang, Y., Mandlekar, A., Yang, Y., Zhu, H., ... & Anandkumar, A. (2022). Minedojo: Building open-ended embodied agents with internet-scale knowledge. _Advances in Neural Information Processing Systems_, _35_, 18343-18362.Advances in Neural Information Processing Systems, 35:18343–18362, 2022.

---

### Meta-Review · Area_Chair_dk2X · 2024-12-20

**Metareview:**

(a) The paper introduces ADAM, an embodied agent that learns causal relationships in the open-world environment of Minecraft. The key finding is that by combining causal discovery methods with LLM-powered embodied exploration, the agent can learn accurate causal graphs without relying on prior knowledge or metadata. The authors claim this leads to robust performance, even in modified environments where pre-trained knowledge about crafting recipes is invalid. The paper demonstrates this through experiments in both original and modified Minecraft environments, showcasing the agent's ability to efficiently learn a causal graph of item dependencies and use it to solve tasks like obtaining diamonds.

(b) Strengths:
- The paper introduces a novel approach by integrating causal discovery with embodied exploration using LLMs in Minecraft.
- The agent's ability to perform well in a modified environment with altered crafting recipes demonstrates the robustness of the causal learning approach and its independence from pre-trained knowledge.
- The experiments provide strong evidence for the agent's efficiency and effectiveness compared to baseline methods.
- The use of a causal graph to represent the agent's learned knowledge enhances interpretability and provides insights into the decision-making process, addressing a common limitation of black-box LLM-based agents.

(c) Weaknesses:
- While the paper proposes a generalizable framework, the experiments are limited to Minecraft. Providing a complementary application in a different environment would strengthen the generalizability claim.
- The paper compares ADAM to text-based baselines, but lacks comparisons to existing multimodal agents, which would provide a more comprehensive evaluation of its performance. Even though in the case that there doesn't exist a comparable baseline, it would have make the paper stronger to create one.

(d) The decision is to accept the paper. Despite some limitations, the paper's novel combination of causal inference with LLM-powered embodied exploration, its strong empirical results demonstrating robust performance, and its contribution to the interpretability of LLM-based agents make it a valuable contribution to the field.

**Additional Comments On Reviewer Discussion:**

- There have been some discussion around disagreements whether comparisons with other agents are sufficient and correct. I think it finally goes down to whether the comparison with OmniJARVIS was correctly presented in the general response. While I agree that the table in the author response appears to be confusing, I think it doesn't fundamentally hurt the paper as the authors do not have the intention to include this in the paper and have stated that they are incomparable.

- One reviewer highlighted the lack of empirical evidence to support the claims of "excellent interpretability" and "close alignment with human gameplay". The authors addressed the interpretability claim by stating that their agent's reliance on a causal graph for knowledge representation and decision-making ensures transparency, unlike methods that rely on learned neural network weights or rule-based knowledge bases.

- Missing Runtime/Memory Analysis: One reviewer pointed out the absence of runtime/memory analysis. The authors responded by stating that all their comparisons utilize the same LLM API (GPT-4-turbo) and that computations are performed externally, not locally. They argued that the overhead of a single step is consistent across all baselines as they conducted their performance experiments on Mineflayer, ensuring uniform evaluation metrics.

With the improvement during the rebuttal phase, the area chair ultimately decided to recommend acceptance.

---

### Decision · Program_Chairs · 2025-01-22

Accept (Poster)